# Current Advances of Atomically Dispersed Metal-Centered Nanozymes for Tumor Diagnosis and Therapy

**DOI:** 10.3390/ijms242115712

**Published:** 2023-10-28

**Authors:** Ruizhen Tian, Yijia Li, Zhengwei Xu, Jiayun Xu, Junqiu Liu

**Affiliations:** 1College of Chemistry and Chemical Engineering, Central South University, Changsha 410083, China; tianrz17@163.com (R.T.); 20223061@hznu.edu.cn (Y.L.); 2Key Laboratory of Organosilicon Chemistry and Material Technology, Ministry of Education, College of Material, Chemistry and Chemical Engineering, Hangzhou Normal University, Hangzhou 311121, China; zhengweixuhznu@163.com (Z.X.); xujiayun@jlu.edu.cn (J.X.)

**Keywords:** single-atom nanozymes, nanocatalytic therapy, cancer diagnosis and therapy, activity modulation, synergistic therapy

## Abstract

Nanozymes, which combine enzyme-like catalytic activity and the biological properties of nanomaterials, have been widely used in biomedical fields. Single-atom nanozymes (SANs) with atomically dispersed metal centers exhibit excellent biological catalytic activity due to the maximization of atomic utilization efficiency, unique metal coordination structures, and metal–support interaction, and their structure–activity relationship can also be clearly investigated. Therefore, they have become an emerging alternative to natural enzymes. This review summarizes the examples of nanocatalytic therapy based on SANs in tumor diagnosis and treatment in recent years, providing an overview of material classification, activity modulation, and therapeutic means. Next, we will delve into the therapeutic mechanism of SNAs in the tumor microenvironment and the advantages of synergistic multiple therapeutic modalities (e.g., chemodynamic therapy, sonodynamic therapy, photothermal therapy, chemotherapy, photodynamic therapy, sonothermal therapy, and gas therapy). Finally, this review proposes the main challenges and prospects for the future development of SANs in cancer diagnosis and therapy.

## 1. Introduction

Cancer is recognized as one of the deadliest diseases threatening human health. Malignant tumors are the second leading cause of human death [1]. Cancer incidence and mortality continue to rise despite rapid advances in medicine and biology. Traditional strategies for clinical oncology treatment, including surgery, radiotherapy, and chemotherapy, still have unavoidable side effects despite their technological maturity and great success [2,3,4,5,6,7,8]. For example, surgery cannot completely cut the tumor tissue, easily leading to cancer recurrence and metastasis. The long-term use of chemotherapeutic drugs will lead to drug resistance in tumor cells, weakening the therapeutic effect. High radiation will cause severe damage to normal tissues and cause great pain to patients [9,10,11]. Therefore, developing new tumor treatment strategies and accurate early cancer diagnosis are significant to cancer prevention and treatment.

With the rapid development of nanotechnology, non-invasive nanocatalytic therapy based on nanozymes is widely used in diagnosing and treating tumors. Nanozymes are based on a nanomaterial with enzyme-like activity and have a kinetic catalytic reaction process similar to that of natural enzymes [12]. Nanozymes have the biological effects of nanomaterials and can also induce apoptosis or ferroptosis of cancer cells by utilizing the particular tumor microenvironment (TME) to generate ROS [13,14,15,16,17]. However, the leakage of metal ions in nanozymes is potentially toxic to normal organs and tissues, limiting their clinical applications [18]. Therefore, maximizing the utilization of metal atoms and improving the catalytic activity of the active site are effective ways to address the toxicity of nanozymes.

With the progress of single atomization and characterization techniques, single-atom nanozymes (SANs) with atomically dispersed active sites, well-defined electronic and geometrical structures, tunable coordination environments, and maximal metal–atom utilization have been developed and exploited [19]. In 2011, Zhang et al. synthesized Pt_1_/FeO_X_ catalysts, and the innovative concept of single-atom catalysis was proposed for the first time, which set off an international research frenzy for single-atom catalysts (SAC) [20]. Subsequently, Dong et al. proposed the concept of SANs and prepared high-performance single-atom catalysts with Fe-N_5_ structure by mimicking the active center of cytochrome P450, whose catalytic rate constant of oxidase-like activity is more than 70 times that of platinum [21]. Since then, various SANs with different mimetic enzyme activities have been developed, including catalase-like (CAT-like), peroxidase-like (POD-like), oxidase-like (OXD-like), and glutathione oxidase-like (GSHOx-like) [22,23,24,25]. The activities of some SANs are even comparable to natural enzymes. SANs with atomically dispersed bimetallic catalytic sites have also been developed. The parallel catalytic effect of the isolated bis-monatomic sites [26,27] and the synergistic effect of the neighboring diatomic pairs [28,29] can further enhance the catalytic performance of the SANs.

The excellent catalytic properties and the structural stability of SANs ensure that they can achieve satisfactory cancer therapeutic effects at relatively low metal concentrations [30]. In addition, a clear structure of the active site is conducive to revealing the conformational relationship and catalytic mechanism to rationally design more suitable SANs [31,32]. SANs can respond to the weak acidity and high levels of hydrogen peroxide (H_2_O_2_) of TME by converting hydrogen peroxide and oxygen into toxic ROS (hydroxyl radicals (•OH), superoxide radicals ($O_{2}^{•-}$), singlet oxygen (^1^O_2_)) while consuming reduced glutathione (GSH) in the tumor cells, thus coming to kill cancer cells [33]. This review mainly introduces the application of SANs-mediated non-invasive nano-catalytic therapy in oncology diagnosis and treatment in recent years, thoroughly exploring the mechanism of the therapy as much as possible. There are three main aspects to improve the effectiveness of SANs in tumor diagnosis and treatment: (1) to enhance the intrinsic catalytic activity of SANs; (2) to enhance the biocompatibility and tumor accumulation ability of SANs; (3) to combine multiple therapeutic modalities (e.g., chemodynamic therapy (CDT), sonodynamic therapy (SDT), photothermal therapy (PTT), chemotherapy (CT), photodynamic therapy (PDT), immunotherapy (IT), sonothermal therapy (STT), gas therapy (GT), and magnetic resonance imaging (MRI)) to synergistically fight tumors (Figure 1).

## 2. Classification of SANs for Cancer Treatment

Recently, various SANs have been used in cancer therapy and have shown excellent anti-tumor activity. Therefore, it is necessary to understand what kinds of SANs for cancer therapy are available, which can help us to understand the current development of this field and have a clearer perception of a more rational design of SANs in the future. This review mainly uses two classification criteria to classify SANs used in oncology therapy, so SANs used in other fields are not discussed in this review.

The first criterion is a classification based on active sites. Firstly, according to the number of active sites, they can be classified into single-atom site nanozymes and dual-atom site nanozymes. Dual-atom site nanozymes are classified into isolated double monoatomic nanozymes and diatomic pair nanozymes based on the size of the distance between the two metal atoms. Isolated double monatomic nanozymes randomly distribute the two metal atoms on the carrier. There is no definite distance range between them. They can seldom be close to each other to form the M1-M2 configuration and more often play a parallel catalytic effect of the two metal active sites. Diatomic pair nanozymes have significant synergistic effects between the metal atoms, which can significantly improve the catalytic activity and selectivity.

The second criterion is categorized according to the carrier. The principle suitable carrier directly affects the coordination environment of the metal atoms, which in turn affects the catalytic activity of SANs. Moreover, the selection of suitable carriers not only enables the construction of SANs suitable for biological applications but also allows SANs to be used as a platform for integrating other therapeutic modalities. Currently, the most widely used are nitrogen-doped carbon carriers, which have good biocompatibility, and their porous structure and surface modifiability are conducive to integrating multiple therapeutic approaches. In addition, the photothermal conversion properties of carbon materials amplify the therapeutic effect of SANs. Other carriers, such as metal oxides and metal–organic frameworks, have played a significant role in tumor therapy.

### 2.1. SANs Classified According to Metal Active Centers

#### 2.1.1. Single-Atom Site Nanozymes

Mn SANs

Manganese (Mn) is an essential element for living organisms. Due to its multivalency and high spin, Mn is a crucial cofactor for many metalloenzymes such as Mn superoxide dismutase (Mn SOD), glutamine synthetase (GS), pyruvate carboxylase and arginase [34,35,36,37]. Different Mn-based nanozymes exhibit CAT-like, SOD-like, and POD-like activities. Manganese-based nanozymes with multi-enzyme activities have been used to regulate nitric oxide levels and ROS levels in mammalian cells [38].

Zhu et al. constructed a manganese-based SAE (Mn/PSAE) by anchoring single-atom Mn to a centrally controlled cubic nitrogen-doped carbon material, which was then constructed by PEGylation (Figure 2a) [39]. The CAT-like activity of Mn/PSAE catalyzes the decomposition of H_2_O_2_ and generates a large amount of O_2_. Subsequently, the OXD-like activity mediates electron transfer to O_2_, which generates abundant cytotoxicity of $O_{2}^{•-}$, which induces apoptosis. In parallel, the POD-like activity of Mn/PSAE catalyzes the decomposition of H_2_O_2_ to produce •OH. These parallel cascade catalytic reactions generate plenty of ROS and effectively kill tumor cells. The amorphous carbon structure of Mn/PSAE exhibits outstanding photothermal properties, with a photothermal conversion efficiency rate significantly higher than that of commercial photothermal reagents. Therefore, Mn/PSAE showed significant therapeutic effects by stimulating multiple ROS production and photothermal activity through the TME.

Cu SANs

In recent years, scientists have found that bacteria exist in almost all tumor sites, and this pathogen–tumor symbiosis system facilitates tumor development, including escaping immune recognition, inhibiting apoptosis, enhancing drug resistance, and inducing distal metastasis [40]. In particular, in treating colorectal cancer, the abundant Fusobacterium (F.) nucleatum will continuously affect the incidence, progression, metastasis, and prognosis of colorectal cancer. Therefore, Wang et al. prepared a protein-based Cu single-atom nanozymes for breaking the pathogen–cancer symbiont to effectively treat colorectal cancer (Figure 2b) [41]. Bovine serum albumin (BSA) endows it with good biocompatibility and high hydrophilicity, making it no apparent toxicity to non-tumor cells (NCM 460 and HK-2 cells) in the 0~60 μg/mL concentration range. The Cu ^+^ in SA-Cu SAN is coordinated with two N atoms on the histidine imidazole ring and two oxygen atoms of aspartic acid and H_2_O. The formed Cu-N bond and Cu-O bond are similar to many natural copper-containing enzymes, showing peroxidase-like activity. Therefore, it can effectively produce •OH, achieve in situ removal of *F. nucleatum*, and restore the increased autophagy level of tumor cells due to *F. nucleatum*. The weak acidity of tumor cells and the overexpression of hydrogen peroxide resulted in better killing selectivity for tumor cells. In addition, BSA-Cu SANs can be entirely cleared by the kidneys, sparing them from long-term systemic toxicity, which is a significant advance for the clinical translation of SANs.

Ir SANs

The search for nanocatalysts with high activity, selectivity, and biological safety profoundly impacts biomedical development. The intrinsic catalytic properties of the central metal, the coordination environment, the metal–carrier interaction, and the metal loading together determine the performance of the nanocatalysts. Whereas the active metal is a double-edged sword, it may undergo complex biophysical–chemical reactions at the nano-biological interface, resulting in potential biotoxicity when exposed to the physiological environment. Therefore, increasing the catalytic activity of metals within a limited metal content is more beneficial for SANs to improve therapeutic efficacy in biomedicine with less toxic side effects.

Iridium (Ir) is a noble metal belonging to the platinum group of elements, chemically stable and highly resistant to corrosion [42]. Ir exhibits excellent catalytic properties due to its multivalency and adsorption capacity for organic substances. Ir (III) complexes have been widely used in tumor therapy in recent years due to their potential chemotherapeutic properties [43,44,45]. Also, due to the ligand and charge transfer potentials favoring the catalytic properties of Ir atoms, scientists have delved into exploring the biomedical significance of Ir-based nanocatalysts [46]. For example, Cheng et al. developed ultra-low metal content Ir SACs (≈0.11 wt%) with fully exposed active sites (Figure 2c) [47]. Ir SACs exhibited significant POD-like activity and high GSHOx-like activity, catalyzing the conversion of endogenous H_2_O_2_ to •OH in weakly acidic TME while eliminating the cytosolic reducer of ROS, GSH. This synergistic interaction between the two enzyme activities increased tumor cells’ significantly elevated ROS levels, leading to further iron death by lipid peroxidation. Ir SAC surface loaded with an anthraquinone ligand-containing GSH-capturing platinum (IV) compound accelerated GSH depletion, exacerbated intracellular redox imbalance, and accelerated tumor cell death.

The formation of protein coronas can hinder the use of nanomaterials in biomedical applications because hard coronas increase the spatial site resistance of active sites and significantly deactivate the performance of nanocatalysts [48]. Nanoparticles encapsulated in erythrocyte membranes hardly adsorb proteins in plasma. Therefore, the erythrocyte membrane can be used as a suitable carrier for intravascular drug delivery, which is expected to reduce the phagocytosis of nanomaterials by the reticuloendothelial system and prolong their circulation time in the body. Finally, the erythrocyte membranes were coated with Pt@IrSACs (Pt@Ir SACs/RBC), which increased the cytocompatibility of the nanoparticles and effectively prevented the formation of a protein canopy. The Pt@IrSACs/RBCs showed excellent therapeutic efficacy and biosafety in a mouse model of triple-negative breast cancer, with complete tumor ablation after a single treatment session and no recurrence. This work provides valuable insights for designing SANs with high performance and biosafety for biomedical applications [47].

Fe SANs

Qin et al. used an interfacial domain-confined coordination strategy to construct Fe single-atom anchored defective carbon dots in poly (ethylene glycol)-modified porous silica nanoreactors (Fe/CDs@PPSNs) [49]. This strategy involves in situ high-temperature carbonization of polymer/nitrogen-containing molecules into nitrogen-doped carbon dots and confinement of interfacial coordination of N and Fe atoms at the interface between the carbon dots and the iron oxide nanoparticles, followed by acid etching in biocompatible porous nanoreactors to remove the excess iron-based nanoparticles (Figure 2d). The Fe- N-C single atoms are formed by strong formation coordination interactions that occur in uniform and connected hierarchical mesopores. The interconnected porous skeleton also facilitates substance transport to expose more active sites, increasing the local reactant concentration and generating ROS more efficiently. Meanwhile, the excellent photothermal conversion efficiency of Fe/CDs@PPSNs further amplifies the effect of the tumor therapy under 808 nm laser irradiation. Moreover, the compositional tunability within the mesoporous pores offers the possibility of encapsulating different metal-based cores to prepare multiple nitrogen-coordinated metal single-atom nanotherapeutics.

Co SANs

Cobalt (Co) is a biologically essential trace element, generally found in vitamin B12, whose central CoIII ion is coordinated to the four nitrogen atoms in the Corin [50]. Various cobalt-based nanozymes show different enzyme-like activities, such as such as CAT-like, SOD-like, OXD-like, and POD-like. Co-N_4_ centers anchored on graphene with N-doped graphene exhibit prominent Fenton-like activity that can produce •OH [51]. Co single atoms on MoS_2_ nanosheets also exhibit remarkable POD-like activity, where Co follows an electron transfer mechanism [52]. Cai et al. constructed nitrogen-doped graphene (Co-SAs@NC) anchored to the active site of Co-N_4_ [53]. Co-SAs@NC’s CAT-like activity breaks down endogenous cellular H_2_O_2_ into O_2_, followed by the conversion of O_2_ into highly cytotoxic $O_{2}^{•-}$ using OXD-like activity, effectively killing tumor cells. By introducing adriamycin, the therapy showed a synergistically enhanced anti-tumor effect.

**Figure 2 ijms-24-15712-f002:**
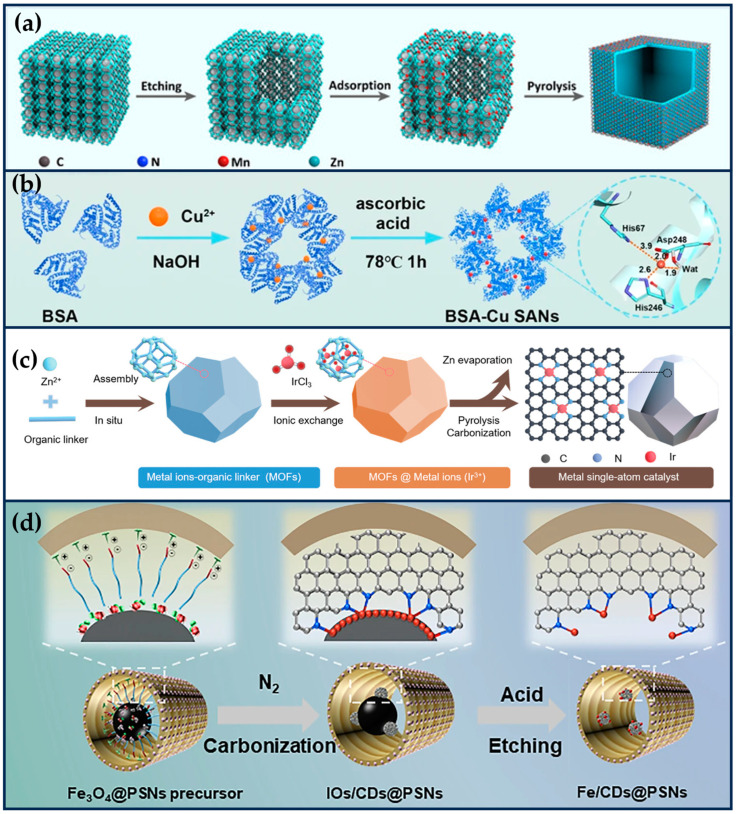
Single-atom site nanozymes. (**a**) The preparation of Mn/SAE (reproduced from [39] with permission from John Wiley and Sons). (**b**) The synthesis of BSA-Cu SAN (reproduced from [41] with permission from Springer Nature). (**c**) The fabrication of Ir1/CN SAC via an ionic exchange strategy (reproduced from [47] with permission from John Wiley and Sons). (**d**) The construction of Fe/CDs@PPSNs by an interfacial-confined coordination strategy (reproduced from [49] with permission from Springer Nature).

#### 2.1.2. Isolated Double Monoatomic Nanozymes

##### Fe/Co Bimetallic SANs

Zhao et al. prepared bimetallic SANs consisting of single-atom Fe and single-atom Co to form non-alloyed isolated atoms (DIA) anchored in N-doped carbon carriers (Figure 3a) [27]. Briefly, Fe ions and Co ions were isolated within the cavities of ZIF-8, and then the surface of ZIF-8 was covered with a Si shell to prevent irreversible fusion and aggregation during high-temperature pyrolysis. Finally, the SiOx shell layer and metal nanoparticles were then etched away using 12% hydrofluoric acid and 1 M hydrochloric acid, respectively, to form FeCo-DIA/NC. Hyaluronic acid (HA) was modified on the surface of FeCo-DIA/NC (FeCo-DIA/NC@HA) to improve its tumor-targeting properties. The Fe-N_4_ active sites in FeCo-DIA/NC mainly exhibit POD-like and OXD-like activities, catalyzing in parallel the formation of •OH and $O_{2}^{•-}$ from H_2_O_2_ and O_2_, respectively. The Co-N_5_ active sites in FeCo-DIA/NC can catalyze in parallel the formation of ^1^O_2_ and $O_{2}^{•-}$ from H_2_O_2_ and O_2_, respectively. The synergistic and “division of labor” bimetallic dual active sites exhibit much higher catalytic activity than single metal sites. The results of in vitro and in vivo experiments showed that FeCo-DIA/NC@HA could significantly reduce cell survival and inhibit tumor growth [27].

##### Cu/Zn Bimetallic SANs

In addition, the catalytic activity of enzyme-like enzymes can be further improved by exploiting the localized surface-isolated exciton resonance (LSPR) effect of metal doping. Liu et al. designed a Cu/Zn bimetallic two-single-atom nanozyme (Cu/PMCS) for treating skin melanoma [54]. The doping of Cu enhanced the photothermal properties, catalytic activity, and GSH depletion capacity of PMCS, and improved the elimination of skin melanoma (Figure 3b) [54]. According to DFT theoretical calculations, Cu doping significantly improves the photothermal performance of PMCS, which is attributed to the introduction of impurity energy levels by Cu ion doping, resulting in a new d-orbital transition with strong spin–orbit coupling. In addition, the negative dielectric effect generated by the transition-induced LSPR enhances the photothermal effect. Under light irradiation, the Cu-based nanomaterials produce hot electrons and holes through the LSPR effect, which may further enhance the GSH depletion capacity and Fenton-like catalytic reaction rate.

These isolated bimetallic two-single-atom nanozymes, possessing higher catalytic activity than monometallic SANs, could significantly increase the ROS level of tumor cells and enhance the anti-tumor effect. However, this is only a superposition of the two metals’ catalytic effect; if more metals are encapsulated in the MOF cavity, it should show better catalytic activity. Pull in the distance between the two metals to construct a diatomic pair of SANs, which can produce the effect of “1 + 1 > 2” by utilizing the synergistic effect between the metals.

#### 2.1.3. Diatomic Pair Nanozymes

Isolated single metal sites usually exhibit end-on conformation for H_2_O_2_ adsorption, making the energy barrier for O-O bond cleavage is higher and unfavorable for forming active intermediates [55]. In nature, some enzymes with binuclear metal sites can accelerate O-O breaking due to their unique geometries and electronic structures, such as cytochrome c oxidase (Fe-Cu hetero-dinuclear active center), methane monooxygenase (Fe-Fe dinuclear active center), and polyphenol oxidase (Cu-Cu dinuclear active center) [56,57,58]. Two adjacent metal atoms as active sites can bind O atoms through double-site adsorption, which facilitates the modulation of D-band centers through electron orbital interactions and further optimizes the adsorption/desorption of oxygen intermediates, accelerating the rate of catalytic reactions [59].

Jiao et al. synthesized FeCuNC nanozymes with Fe-Cu heteronuclear active sites using a template-assisted method (Figure 3c) [28]. The maximum reaction rate of FeCuNC was 7.66 times higher than that of FeNC, which was attributed to the ability of the bimetallic sites of FeCuNC to significantly reduce the activation energy barrier of H_2_O_2_ (16.49 kJ/mol for FeCuNC and 29.63 kJ/mol for FeNC). In addition, the O-O bond of H_2_O_2_ was more susceptible to dissociation due to the higher dissociative adsorption energy of the FeCu heteronuclear site for H_2_O_2_. The intrinsic photothermal performance of FeCuNC nanozymes (52.52% photothermal conversion efficiency at 808 nm) further improved the POD-like activity. For specific recognition of sentinel lymph nodes (SLN) and tumors, FeCuNC nanozymes were coated with 4T1-Luc cell membranes, which effectively inhibited tumor metastasis and recurrence by upregulating the expression of TNF-α and downregulating the expression of IL-10. This work provides implications for designing high-performance nano-catalytic therapeutic agents and new ideas for inhibiting tumor lymphatic metastasis [28].

### 2.2. SANs Classified According to Carrier

#### 2.2.1. Nitrogen-Doped Carbon-Based SANs

Nitrogen-doped carbon nanomaterials are a highly desirable carrier for constructing SANs because of their high carrier density, high catalytic activity, and high chemical affinity. The coordination of nitrogen with various metals (noble and non-noble) forms a unique electronic structure, and the synthesis process is often accompanied by the generation of topological defects, which makes the constructed SANs exhibit superior catalytic activity. In addition, nitrogen-doped carbon nanomaterials have tunable morphology and ordered porosity properties, allowing them to function as platforms for integrating other therapeutic modalities to facilitate tumor therapy.

Lu et al. prepared biomimetic hollow nitrogen-doped carbon spheres doped with single-atom copper species (Cu-HNCS), which can directly catalyze the decomposition of O_2_ and H_2_O_2_ to $O_{2}^{•-}$ and •OH in acidic TME (Figure 4a) [60]. The Fenton reaction conversion rate of Cu-HNCS is about 5000 times higher than that of commercial Fe_3_O_4_ nanoparticles. DFT calculations reveal that the high catalytic activity of the Cu-N_4_ sites is mainly attributed to the low apparent energy barriers of the catalytic reaction and the negative Gibbs free energy. The low metal loading and high tumor inhibition rate of Cu-HNCS suggest an excellent potential for clinical applications.

#### 2.2.2. MOF-Based SANs

Artificially creating specific coordination sites within the framework of MOFs for anchoring heterogeneous metal atoms without loss of crystallinity and porosity is also a viable strategy for constructing SANs. For example, Yaghi et al. reported a single-atom catalyst in which a single Cu atom anchored to an oxygen atom of the -OH/-OH_2_ species covering the defective sites of the Zr oxide cluster of MOF UiO-66, achieving 100% selective CO oxidation [64].

Wang et al. anchored Pt nanoparticles and single Pt atoms to the skeleton of the Prussian blue analog Mn[Co(CN)_6_] _2/3_
_1/3_ MOF (MC_2/3_, represents one-third of the internal missing linker) through the “missing-linker-confined coordination” strategy, and prepared traditional nanozymes (MC_2/3_CpNE) and single-atom nanozymes (MC_2/3_Cp-SAN) [65]. Specifically, the biocompatible polyvinylpyrrolidone polymer (PVP) and Mn ions were mixed to form a homogeneous solution, and the K_3_[Co(CN)_6_] solution was added to form a milky white colloidal suspension, which was the self-assembled MC_2/3_. MC_2/3_ loaded with Ce6 and anchored single Pt atoms (MC_2/3_Cp-SAN) can be obtained by adding the photosensitizer Ce6 and PtCl_2_ during the formation of MC_2/3_, with loadings of 27.8 wt% and 8.48 wt%, respectively. MC_2/3_Cp-SAN exhibited stronger CAT-like activity together with cyclic catalytic stability than MC_2/3_CpNE. The in vitro and in vivo results demonstrated that MC_2/3_Cp-SAN could efficiently catalyze the conversion of excess H_2_O_2_ to O_2_ in tumor cells, thereby enhancing the photodynamic therapy (PDT) efficiency for cancer cells. This work not only proves the superiority of SANs over traditional nanozymes but also broadens the strategy for synthesizing SANs based on MOFs.

Hypoxia of solid tumors affects the therapeutic efficacy of PDT. Wang et al. used Mn_3_[Co(CN)_6_]_2_ MOF material as a carrier and doped Ru into the framework to form a single-atom catalytic site by substituting part of Co (OxgeMCC-r SAE) (Figure 4b) [61]. The loading of single-atom Ru is as high as 2.23 wt%, and it can exhibit excellent CAT-like activity for converting endogenous H_2_O_2_ to O_2_, which enhanced the ^1^O_2_ production of the photosensitizer chlorin (Ce6) encapsulated in the framework. At the same time, OxgeMCC-r SAE also enables T1-weighted MR imaging for in vivo tracking of therapeutic agents due to the presence of high-spin Mn-N_6_ (S = 5/2) species.

#### 2.2.3. Carbon Dots-Based SANs

Carbon dots (CDs), as graphene derivatives, have uniform and ultra-small sizes (<10 nm) [66,67,68]. The CD surface is rich in coordination unsaturated chemical groups (-OH, -CO, and -NH_2_), which can anchor the active metal sites and catalyze various biochemical reactions [69]. The high specific surface area of the CDs ensures complete contact between the active sites and the substrates, which enhances the catalytic activity.

Wang et al. used biocompatible carbon dots as carrier materials loaded with Ru single-atoms to prepare Ru SANs with multiple enzyme-like activities and stability [70]. The OXD-like, POD-like, and GSHOx-like activities of Ru SANs can synchronously catalyze the production of ROS and the depletion of GSH, which can amplify the ROS damage and ultimately lead to the death of the cancer cells. The specific activity of the POD-like activity of the Ru SAEs (7.5 U/mg) is 20 times higher than that of Ru/C. The theoretical results suggest that the electron transfer from the 4D orbital of the Ru single atom to the O atom in H_2_O_2_ effectively activates H_2_O_2_ to produce •OH, thus exhibiting excellent catalytic activity.

Glioblastoma (GBM) is a fatal recurrent brain tumor for which there is no complete cure. Muhammad et al. explored SANs-mediated catalytic therapy for precisely targeting drug-resistant GBM (Figure 4c) [62]. Microwave-assisted pyrolysis was employed to prepare ultrasmall carbon dots (Fe-CDs) loaded with Fe single atoms, using Gerbera leaves extracts as C and N sources, exhibiting six mimetic enzyme activities, including OXD-like, CAT-like, SOD-like, POD-like, GPx-like, and thiol-like peroxidases (TPx), which are responsible for regulating intracellular redox homeostasis. Surface modification of Fe-CDs with angiopep-2 (Fe-CDs@Ang), a BBB permeable and glioma-targeting peptide, enabled GBM-targeted delivery across the BBB. Fe-CDs@Ang successfully targeted LRP-1, which is highly expressed by brain capillary endothelial cells and tumor cells and was able to be Fe-CDs@Ang aggregated in acidic lysosomes (pH 4~5), exhibited OXD/POD-like activity, and disrupted the lysosomal degradation ability, thereby activating autophagic flow. In addition, Fe-CDs @ Ang possessed SOD, CAT, and GPx-like activities, which acted as ROS modulators to enhance autophagy and lysosome-based apoptosis. The cascade enzymatic activity of Fe-CDs stimulated autophagy and effectively inhibited the growth of drug-resistant tumors. The green, drug-free nanomedicine therapeutic strategy provides a paradigm for developing novel biomimetic nanozymes.

#### 2.2.4. Metal Oxides-Based SANs

Titanium dioxide (TiO_2_) nanoparticles are a typical class of inorganic sonosensitizers that produce ROS by separating electrons (e^−^) and holes (h^+^) from the energy band structure through ultrasonic excitation. Transition metal ion-doped sonosensitizer can significantly increase the separation of e^−^ and h^+^ and inhibit their complexation, enhancing ROS production efficiency [71]. In addition, the Fenton reaction is also an efficient way to generate ROS. However, the optimum pH of the Fe^2+^-mediated Fenton reaction is 2–4, whereas copper-based Fenton nanocatalysts are more efficient in weakly acidic TME. Breast cancer, the most common cancer worldwide, is one of the major public health problems threatening women’s health, and CT is currently the mainstream strategy to inhibit the progression of TNBC, but it is limited by poor efficacy and serious side effects [72,73]. SDT is an effective treatment for breast cancer, which uses the massive production of ROS to induce oxidative stress damage and trigger apoptosis and necrosis of tumor cells. However, mutant breast cancers, especially TNBC, have evolved specific antioxidant defenses that limit the killing efficiency of SDT [74,75]. Chen et al. rationally constructed a single-atom Cu-doped hollow TiO_2_ nano-sonosensitizer (Cu/TiO_2_) for the synergistic treatment of TNBC using SDT and CDT (Figure 4d) [63]. The single atom of Cu occupies the most stable Ti vacancy in the hollow TiO_2_, which not only enhances the catalytic activity of the Cu-mediated Fenton-like reaction but also significantly improves the ROS generation efficiency by promoting the e^−^ and h^+^ separation, which significantly improves the acoustic dynamic efficiency of TiO_2_ and increases the efficiency of ROS generation. In vivo experimental results showed that the Cu/TiO_2_-PEG nano-sonosensitizer achieved a combined tumor inhibition rate of 70.35%. The acoustic/chemical synergistic therapy based on single-atom-doped nano-sonosensitizers provides a new paradigm for the non-invasive treatment of refractory breast tumors.

Compositionally tunable two-dimensional (2D) MoS_2_ nanosheets, as an inorganic co-catalyst, are ideally suited to act as SAC nanocarriers due to their large specific surface area, unique planar structure, and abundant active sites [76]. Yang et al. dispersed single-atom Fe sites on MoS_2_ nanosheets sulfur-rich vacancies and active Mo^4+^ sites to construct a 2D composite nanocatalyst for achieving co-catalytic synergistic therapy of tumors [77]. The therapy S vacancies can increase the surface electron density and promote the electron capture by H_2_O_2_ to generate •OH. The reductive Mo^4+^ sites can accelerate the conversion of Fe^3+^ to Fe^2+^. Meanwhile, the 2D lamellar structure can more fully expose Fe atoms, sulfur vacancies, and Mo^4+^ active sites, which facilitates the synergistic overall reaction process and improves the POD-like activity (*V*_max_ = 4.37 × 10^−8^ M s^−1^, *K*m = 15.06 μM). In addition, in vitro and in vivo experiments demonstrated that the constructed nanocatalysts have promising therapeutic efficacy and biosafety, and the concept of co-catalysis will be beneficial for developing nanocatalysis in tumor therapy.

Zhou et al. constructed a polyvinylpyrrolidone (PVP)-modified Cu single-atom nanozyme MoOx-Cu-Cys-PVP (MCCP SANs) using a coordination-driven self-assembly strategy [22]. The cRGDfc peptide capable of recognizing α_v_β_3_ integrins was next loaded on the MCCP SANs surface, allowing it to target tumors actively. Cu single atoms self-assembled on the MoOx NPs surface driven by amino acid (L-cysteine) binding at a tunable content ranging from 1.59 to 26.67 wt%. At 10.10 wt% Cu atoms, the MCCP SANs exhibited the best CAT-like activity, 138 times that of the MnO_2_ nanozymes and 14.3 times the affinity for the substrate than that of the natural catalase. The weak metal–support interactions between the Cu and O atoms in MoOx formed L-Cys-Cu· · ·O electronically structured active sites on MCCP. The amino acid bridge accelerates the transfer of electrons from individual Cu atoms to O in the MoOx carrier, which endows MCCP with an active site similar to that of catalase as well as optimal adsorption energy. Meanwhile, MCCP can react with H_2_O_2_ in TME to produce highly toxic •OH via a Fenton-like reaction, while X-ray irradiation converts locally elevated O_2_ to singlet oxygen (^1^O_2_), which catalyzes the enhancement of apoptosis in tumor cells explicitly. In addition, MCCP has a strong tumor penetration ability and can be gradually removed from tissues, thus ensuring no damage to normal tissues. This novel way of constructing SANs provides new ideas for SANs in biomedicine [22].

## 3. Modulation of Activity of SANs for Cancer Therapy

SANs can generate large amounts of ROS to kill tumor cells in response to the weak acidity and high H_2_O_2_ levels of the TME. Different types of SANs achieve the regulation of intracellular ROS levels mainly through POD-like, OXD-like, CAT-like, and GSHOx-like activities to kill cancer cells. Among them, the POD-like activities of SANs catalyze the generation of •OH with H_2_O_2_ as the electron acceptor when functioning. Due to the high reactivity of •OH, which can use various small molecules and macromolecules such as nucleic acids, lipids, and proteins as electron donors, causing severe oxidative damage to cancer cells. The CAT-like activity of SANs can decompose the endogenous H_2_O_2_ of the cancer cells into O_2_, which alleviates the anoxic environment of the tumor, thus increasing the efficiency of the generation of ^1^O_2_ in RT, PDT, and SDT. The OXD-like activity of SANs can catalyze the generation of H_2_O_2_ and $O_{2}^{•-}$, whose cancer cell-killing ability is weaker than that of •OH and needs to be combined with other enzyme-like activities to kill tumors. The GSHOx-like activity of SANs can consume excessive intracellular GSH, thus weakening the antioxidant ability of cancer cells and playing a very auxiliary therapeutic effect.

In order to further improve the anti-tumor ability of SANs, maximizing their catalytic activity is an effective strategy. This review mainly outlines two directions to improve the performance of SANs. The first direction is to improve the intrinsic enzyme-like activity of SANs, most notably the metal active center, and optimize the metal center coordination environment. The second direction is to improve the TME in which SANs reside. Tumor cells also suffer from hypoxia and overexpression of GSH, which reduce the level of ROS produced by SANs and weaken the effect of tumor therapy. Therefore, improving the anoxic environment of the tumor, depleting the over-expressed GSH, and increasing the concentration of H_2_O_2_ can maximally return the catalytic therapeutic performance of SANs.

### 3.1. Regulation of Intrinsic Enzyme-like Activities of SANs

#### 3.1.1. Defect Engineering

Defect engineering can effectively regulate the geometry and electronic structure of single-atom catalysts and endow the catalysts with unique physicochemical properties, which is an effective strategy to modulate the catalytic activity. Common approaches include heteroatom doping, modulation of vacancies, and introduction of topological defects [78]. The properties of metal catalytic sites in intrinsic defects (e.g., edges) are different from that of the basal plane, which can profoundly affect the catalytic activity of SANs due to their unique geometrical and electronic structures that generate different local electronic environments [79]. In addition, edge sites allow substrates to fully contact the active center, speeding up mass transfer and accelerating reaction kinetics.

Kim et al. developed edge-rich Fe-based SANs (FeNC-edge), in which part of the Fe atoms are anchored to H_2_O_2_-mediated edge sites in N-doped mesoporous carbon nanoparticles (NC), which were synthesized by polymerization reaction using phenol and melamine as carbon and nitrogen sources, respectively [80]. H_2_O_2_ was decomposed into OH- during the hydrothermal process at 180 °C, which could lead to the generation of abundant edge sites in NC. The generated edge sites can effectively immobilize Fe atoms due to the lower work function of the frontal edge sites and lower FeN_4_ edge formation energy. After adsorption of the Fe precursor to the carbon matrix, the FeNC-edge was obtained after annealing at 900 °C under an argon atmosphere. Density functional theory (DFT) calculations showed that the introduction of a large number of edge sites increased the electron density of the edge Fe active center and facilitated the electron transfer, resulting in more excellent POD-like, OXD-like, and CAT-like activities of FeNC-edge (Figure 5a). The second-order apparent constants of the POD-like and OXD-like activities of FeNC-edge were 9.0 and 1.3 times higher than those of FeNC, respectively. The in vitro and in vivo results further indicated that FeNC-edge could effectively convert H_2_O_2_ and O_2_ into •OH and $O_{2}^{•-}$, disturb the redox balance of cells, accelerate the apoptosis of tumor cells, and exhibit excellent therapeutic effects on tumors.

Despite the excellent performance of FeNC-edge, another issue to be considered is that the introduction of the edge structure also increases the Fe content of the material from 0.45 wt% to 0.72 wt%, so whether the increase in activity is due to the increase in the number of active sites or to the edge structure needs to be further considered. As can be seen in the preparation of the materials section of the article, the preparation of FeNC was only completed with less H_2_O_2_ etching than the preparation of FeNC-edge, which means that the authors did not deliberately control the consistency of the Fe content in the two materials when comparing the activities. Therefore, to more fully illustrate the effectiveness of the edge structure, it is necessary to compare the effect of the edge structure’s presence or absence on the materials’ catalytic activity in the presence of a consistent Fe content. Revealing the relationship between enzyme activity and structural properties is essential for guiding the synthesis of SANs and developing high-performance artificial enzymes for tumor therapy.

#### 3.1.2. Regulation of the Coordination Environment

Nanozymes are a class of nanomaterials with intrinsic mimetic enzyme activity, which have excellent catalytic activity and physicochemical properties. Meanwhile, compared with similar materials, nanozymes have the advantages of low cost, excellent stability, and adjustable catalytic activity, and they are widely used in the treatment and diagnosis of tumors. However, the catalytic activity of nanozymes is still far from that of natural enzymes, while the disadvantages of poor substrate selectivity, low atom utilization efficiency, and intrinsic biotoxicity have hindered their further application. Due to their well-defined electronic structure, excellent substrate selectivity, and maximum atomic utilization efficiency, the new generation of SAN has been explored for biomedical applications. Further enhancement of the catalytic activity of SANs by modulating the coordination environment of metal atoms is an effective way and essential for its further applications.

##### Regulating the Number of Coordination N

The MN_4_-type SANs with non-polar coordination structures have a symmetrical electron distribution that limits their adsorption capacity and catalytic activity. Therefore, changing the electron distribution of SANs by adjusting the number of coordinated N atoms is an effective strategy to improve its catalytic activity. By theoretical calculations, Wang et al. have designed a series of non-homogeneous molybdenum single-atom nanozymes (MoSA-Nx-C, x = 2, 3, 4) [82]. The molybdenum site coordination number has a strong correlation with the POD-like specificity. MoSA-N_3_-C exhibits a dedicated POD-like activity and achieves this behavior through a homocleavage pathway, whereas the MoSA-N_2_-C and MoSA-N_4_-C catalysts have different heterocleavage pathways. This coordination number-dependent enzyme specificity is attributed to the geometrical differences and the orientation of the front molecular orbitals of MoSA-Nx-C.

HRP has a heme moiety centered on active iron and the imidazole nitrogen of histidine as the fifth axial ligand [83]. The axial nitrogen ligand plays a crucial role in stabilizing the active structure and enhancing enzyme activity. In chemical catalysis, the axially coordinated MN_5_ structure is capable of downgrading the reaction activation energy and increasing the catalytic rate. Xu et al. synthesized nanozymes (FeN_5_ SAN) with atomically dispersed FeN_5_ structures by a melamine-mediated two-step pyrolysis strategy [81]. The axially coordinated N optimized the electronic structure of the active site and reduced the activation energy of H_2_O_2_ (Figure 5b). Accordingly, FeN_5_ SAN exhibited stronger POD-like activity than FeN_4_ SAN, with a second-order apparent constant 7.64 times higher than that of FeN_4_ SAN. In the treatment of 4T1-bearing mice, FeN_5_ SAN effectively killed tumor cells.

##### Heteroatom Doping

Ferroptosis is a cell death pathway that accumulates lipid peroxides in an iron-dependent manner. Scientists have employed different iron-based nanozymes to induce ferroptosis by oxidizing polyunsaturated fatty acids (PUFAs) and LPO [84]. However, the efficiency of ferroptosis stress induced by iron-based nanozymes is limited due to the special tumor microenvironment, mainly caused by the deficiency of Fenton activity and the overexpression of GSH. To overcome this limitation, Zhu et al. prepared nickel single-atom nanozymes (S-N/Ni PSAN) enriched with marginal sulfur (S) and nitrogen (N) modifications by a strategy of anion exchange from the viewpoint of improving the intrinsic activity of single-atom nanozymes [23]. NiCo Prussian blue analog (NiCo PBA) was first used as a precursor, and then NiS nanocubes were prepared by etching Co atoms using Na_2_S via anion exchange. Then, a polydopamine (PDA) layer was coated on the surface of NiS nanocubes, and the polydopamine layer was pyrolyzed to a nitrogen-doped carbon (N-C) backbone by high-temperature pyrolysis. During calcination, volatile S species (bp 444.7 °C) are more inclined to the edge-selectively sulfated N-C backbone, anchoring Ni atoms to the edge S-rich N-C skeleton. S species are essential components of many natural enzymes and critical in transferring electrons from the substrate to the enzyme active center. The vacancies and defective sites of the nitrogen sulfide atoms allow S-N/N/Ni PSAE to exhibit stronger POD-like activity and GSHOx-like activity than N/Ni PSAE. For POD-like activity, the SA value of S-N/Ni PSAE (115 U/mg) was 11.6 times higher than that of N/Ni PSAE (9.9 U/mg). S-N/NiPSAE could produce •OH and consume GSH more efficiently than N/Ni PSAE in ferroptosis-based tumor therapy, inducing GPx-4 inactivation and irreversible LPO, leading to ferroptosis of tumor cells and better inhibition of tumor growth. This work enhances the catalytic effect of multi-heteroatom doping and provides an effective strategy for ferroptosis-based anti-tumor methods [23].

By simulating the active center of horseradish peroxidase, Zhu and Li’s group designed SANs with Fe-N_3_S and Fe-N_3_P structures, respectively, and the maximum reaction rate of Fe-N_3_S was 2.10 times that of Fe-N_4_ when H_2_O_2_ was used as a substrate, and the activity of Fe-N_3_P was even comparable to that of the natural enzyme. This is mainly because S and P are less electronegative than N, which can act as electron donors so that electrons can be transferred from S/P → Fe → N, lowering the potential barrier for O formation on the surface of the nanozymes and leading to faster kinetics (Figure 5c) whereas the difference in electronegativity between S and P may have led to the difference in activity between the two structural nanozymes [85,86].

#### 3.1.3. X-ray Irradiation

For the purpose of enhancing the enzymatic activity of SANs, in addition to maximizing the atomic utilization by increasing the surface area, introducing external fields into the catalytic process to improve the catalytic efficiency is also a proven strategy. For example, under X-ray irradiation, the conversion of Cu II species in copper-based nanoparticles to Cu I is accelerated, thus enabling faster conversion of H_2_O_2_ to •OH [87]. Also, since X-rays do not have a depth limitation, such a strategy could be applied to treating deep tumors.

Given the similar kinetics between the conversion processes of Cu I/Cu II and Fe II/Fe III, Zhu et al. introduced X-rays into the catalytic reaction system of iron-based SANs (FeN_4_-SAN) to increase the enzyme activity of Fe II/Fe III by accelerating the rate of its conversion, which is a decisive step in the generation of ROS [88]. Thus, FeN_4_-SAzyme can produce -OH faster to consume GSH. To solve the problem of H_2_O_2_ insufficiency, FeN_4_-SAN was compounded with natural glucose oxidase (GOD) to obtain a therapeutic agent with H_2_O_2_ self-supply ability, which ensured the continuous production of •OH and enhanced in situ apoptosis and ferroptosis. This external field-enhanced SANs catalysis paradigm can be extended to various external fields and other nanozymes to enhance enzyme activity.

#### 3.1.4. Regulation of the Central Metal Atom

Recently, introducing secondary metal atoms can effectively improve the catalytic activity of SACs. DSACs can modulate the d-band center through the entanglement of electronic orbitals, which can effectively optimize the adsorption process of intermediates on the active site [89]. Therefore, DSACs can effectively improve intrinsic activity and selectivity. For example, introducing a single Cu site in Pd SACs transfers part of the density state of Pd to the Fermi energy level. Meanwhile, introducing Cu sites can also modulate the d-2π* coupling between Pd and adsorbed N_2_, thus improving N_2_ chemisorption and protonation, and delaying hydrogen precipitation. Ultimately, the Faraday efficiency and NH_3_ synthesis efficiency of Pd-Cu DSACs were higher than those of Pd SACs [90]. It was also shown that the synergistic effect between Co and Ni sites led to a significant increase in the adsorption and desorption efficiencies and a decrease in the reaction thresholds of the monodispersed Co-Ni dinuclear catalysts [91]. Similarly, monoatomic Pt sites can modulate the electronic environment of the adjacent Fe sites, thereby optimizing the oxygen reduction catalytic activity [92]. Recently, scientists have also explored the role of DSACs in tumor therapy.

Extension of SACs from single-atom to dual-atom sites is an effective method to enhance catalytic activity. Wang et al. prepared atomically dispersed Fe,Pt binuclear nanozymes ((Fe,Pt)SA-N-C) with a distance of 2.38 Å between Fe_1_ (Fe-N_3_) and Pt_1_ (Pt-N_4_) by employing a secondary doping strategy [26]. DFT calculations showed that the isolated Pt sites can modulate the 3D electronic orbitals of Fe-N_3_ and enhance the activity of H_2_O_2_ and the adsorption of Fenton-like reaction intermediates on Fe-N_3_ (Figure 5d). Thus, the activity out of (Fe,Pt)SA-N-C was enhanced by about 30% over the single Fe site. After PEG modification, (Fe,Pt)SA-N-C-FA-PEG was able to effectively catalyze the production of large amounts of toxic •OH from H_2_O_2_ in the TME and induce apoptosis of tumor cells. In addition, based on the graphitized carbon matrix, (Fe,Pt)SA-N-C-FA-PEG exhibited a strong photothermal conversion efficiency (36.8%) under the irradiation of 808 nm laser light, which improved the tumor inhibition efficiency without any significant damage to major organs.

### 3.2. Ameliorating the TME

Solid tumors are hypoxic, and the level of H_2_O_2_ in TME is usually below a certain threshold (≈100 × 10^−6^ M) [93]. Increasing the content of O_2_ and H_2_O_2_ in the tumor site can further improve the catalytic therapeutic effect. NADH oxidase catalyzes the reaction between O_2_ and NADH to produce NAD^+^ and H_2_O_2_, which plays a vital role in regulating cellular redox and maintaining normal cell growth. The NADH/NAD^+^ redox pair acts as an electron transporter, providing protons to the mitochondrial electron transport chain (ETC), and is also required for adenosine triphosphate (ATP) production in tumor glycolysis and oxidative phosphorylation metabolism. Therefore, disrupting the NADH/NAD^+^ balance not only effectively inhibits ETC and reduces mitochondrial aerobic respiration, but also interferes with the metabolic processes of tumor cells and reduces their ability to generate ATP [94,95].

Liu et al. encapsulated Ir(acac)_3_ in ZIF-8 (Ir(acac)_3_@MOF), and then, nitrogen-rich melamine and dried Ir(acac)_3_@MOF were calcined at high temperatures to obtain IrN_5_ SAN with axial N coordination [25]. IrN_5_ SAN has an asymmetric electron distribution and exhibits OXD-like, POD-like, CAT-like, and NOX-like properties. The synergistic interaction between the central metal Ir atom of IrN_5_ SAN and the axial N-coordinated structure efficiently optimizes the free energies of the various transition states on IrN_5_ SAN, which exhibits a better enzyme-like catalytic activity than that of IrN_4_ SAN. IrN_5_ SAN could catalyze the decomposition of H_2_O_2_ to O_2_ at the tumor site, effectively alleviating tumor hypoxia. Meanwhile, IrN_5_ SAN could mimic NOX to catalyze the generation of H_2_O_2_ from NADH and inhibit mitochondrial ETC and aerobic respiration. The increased O_2_ and H_2_O_2_ at the tumor site could enhance the OXD-like and POD-like activities of IrN_5_ SAN, increase the level of ROS in tumor cells, and cause irreversible oxidative damage. The decrease in NADH would break the NADH/NAD^+^ balance and inhibit the ATP produced by tumor cells through the metabolic processes of glycolysis and oxidative phosphorylation. Even though IrN_5_ SAN disrupts the normal energy metabolism of tumor cells, tumor cells can use fatty acid oxidation (FAO) to survive in nutrient-deficient environments through metabolic reprogramming (Figure 6a). Therefore, the attenuated FAO metabolism of tumor cells, ceruloplasmin (Cer), was further loaded into IrN_5_ SAN. Cer, as a fatty acid synthase inhibitor, could effectively inhibit phospholipid synthesis and lipid remodeling in tumor cells and reduce the migration and invasive ability of tumor cells. In order to improve the biocompatibility of IrN_5_ SAN, the surface of IrN_5_ SAN was further modified with a trithiol-terminated polymethacrylic acid. IrN_5_ SAN/Cer significantly enhanced tumor therapy’s therapeutic effect by destroying the tumor redox and metabolic balance by mimicking the enzyme cascade reaction. With the dual function of breaking redox and metabolic homeostasis, this SAN provides a new perspective for nano-catalytic therapy [25]. One problem is that the reaction catalyzed by NOX is an oxygen-consuming process, and it is theoretically debatable whether the oxygen content of tumor cells can be increased even though aerobic respiration in mitochondria is inhibited.

SANs with POD-like activity can alter cellular redox balance and hold great promise for tumor therapy. However, the “cold” immune microenvironment and limited H_2_O_2_ in solid tumors severely limit its efficacy. Zhu et al. designed a light-controlled oxidative stress amplifier system that co-encapsulates Pd-C SANs and camptothecin in an agarose hydrogel to enhance synergistic anti-tumor activity by self-producing H_2_O_2_ and transforming “cold” tumors [96]. In this nano-enzymatic hydrogel system, the Pd-C SANs convert near-infrared laser light into heat, leading to the degradation of the agarose, which releases the camptothecin. The camptothecin increases H_2_O_2_ levels in the tumor by activating nicotinamide adenine dinucleotide phosphate oxidase, which improves the catalytic properties of SANs with POD-like activity (Figure 6b). In addition, the combination of PTT, CT, and SANs-based catalysis further promotes tumor immunogenic death and enhances anti-tumor immunity. The findings reveal the synergistic anti-tumor potential of a novel SAN/CT drug-based hydrogel system.

### 3.3. Increasing the Specific Surface Area of SANs

Employing SANs-based ROS generators has become an effective strategy for mediating tumor therapy, but in physiological environments, problems such as biomass adsorption and micropore clogging occur, severely affecting substance transport and reducing the number of available active sites. Even if ROS are generated, they cannot diffuse out to act, owing to the limitations of the half-life time (<200 ns) and diffusion distance (≈20 nm) [97]. Here, modulating the morphology of SANs to increase the substrate accessibility as well as the diffusibility of ROS will further increase the tumor therapeutic effect of SANs.

Liang et al. encapsulated metalloproteins as single-atom templates in ZIF-8 and pyrolyzed them to obtain highly active Fe-centered SANs, which could regulate the mesopore size and the coordination environment of the metal-active centers during the carbonization process [98]. The POD activity of Fe-SANs was 25, 23, 47, and 1900 times that of natural enzymes, nanozymes with metal ions as single-atom sources, nanozymes with large mesopores induced by Zn evaporation, and HRP enzymes immobilized in ZIF-8, respectively. However, the disadvantage of this approach lies in the higher price of ferritin, which increases the cost of making SANs.

Xing et al. exploited the “PDA-assisted morphology fragmentation” strategy to prepare MOF-derived SANs (C-NFs) with unique flower-like structures, which significantly improved the accessibility of the active sites (Figure 7a) [99]. The polyphenol oxidase-like activity of the zeolite imidazolium backbone can catalyze the in situ polymerization of dopamine, leading to a change in the MOF morphology and inducing the cross-linking of 2D nanosheets into 3D nanoflower structures. The modification of PDA led to the formation of a large number of open micropores and defective mesopores (≈4 nm) in the petal-like lamellae of the C-NFs, a large pore spacing (≈39 nm), a high specific surface area (388 m^2^g^−1^), and an ultra-high metal loading (27.3 wt%) significantly enhanced the POD-like activity and the ROS-generating ability of the C-NFs. Horse spleen desferrin coated with DOX effectively penetrated into the pores of C-NFs (interpetal pores ≈ 39 nm), improving the water dispersion of C-NFs while combining with CT to kill tumor cells. The catalytic production of •OH in the TME induced effective oxidative stress, reduced drug resistance, and improved cell sensitivity to CT.

### 3.4. Other Approaches to Regulating the Activity of SANs

Surface-modified DNA molecules modulate the activity of SANs. To cope with the low activity and uncontrollability of ferroptosis inducers, Cao et al. prepared an adaptive ferroptosis platform based on DNA-modified Fe SANs (macDNAFe/PMCS SANs) (Figure 7b) [100]. PMCS SANs exhibited high OXD-like and POD-like activities, which were 70-fold and 50-fold higher than those of the Fe-based nanozymes, respectively, and GSH depletion capacity. Through van der Waals interactions, C-rich monolayer DNA (cDNA), ATP aptamer (aDNA), and MUC-1 aptamer (mDNA) were encapsulated on the surface of Fe/PMCS SANs, which enhanced the affinity of Fe/PMCS SANs for H_2_O_2_ and substrates and accelerated the generation of ROS. Meanwhile, mDNA increased the affinity of Fe/PMCS SANs for cancer cells and enhanced the selective killing of cancer cells. Subsequently, overexpression of ATP (100~500 × 10^−6^ M) and lysosomal acidity (pH ≈ 5) in tumor cells could remove the aDNA and cDNA shielding effects, exposing the active site of SANs and selectively removing GSH from tumor cells. macDNAFe/PMCS SANs selectively enhanced cancer cell ferroptosis in mouse colon and breast cancer models, showing prominent therapeutic effects. Integrating responsive molecules with SANs provides new insights for preparing iron death inducers with high activity, controllability, and selectivity.

## 4. Synergistic Treatment of SANs with Other Therapies

SANs have the ability to catalyze the generation of ROS efficiently and are also able to remain stable over a wide range of temperatures and pH, as well as higher ionic strengths compared to natural enzymes. In addition, many SANs are significantly more active in weakly acidic environments than in neutral environments, thus enabling specific treatment of the tumor microenvironment (TME). Moreover, the concept of parallel or cascade catalysis allows SANs to simultaneously catalyze H_2_O_2_ and O_2_ to produce •OH and $O_{2}^{•-}$, respectively. Simultaneously, SANs are an exceptional integrative platform for combining multiple therapeutic modalities for synergistic efficacy to maximize tumor eradication. This section describes specifically the current paradigms of combination therapy with SANs and delves into the mechanisms of their synergistic treatment.

### 4.1. Photothermal Therapy (PTT)

General photothermal therapy (60–80 °C) (PTT) is prone to cause damage to normal tissues, while low temperature (40~45 °C) PTT can effectively destroy the primary tumor, avoiding thermal damage to normal tissues and unnecessary inflammation/immunosuppression. It also improves the effect of the corresponding combination therapy and shows higher anti-tumor effects and lower side effects in clinical practice [101,102]. Li et al. mixed Cu/Mn-BTC MOF precursors, which provide ordered and periodic heteronuclear bimetallic nodes, with dicyandiamide, and then pyrolyzed at high temperatures in argon to obtain nitrogen-doped carbon nanosheets enriched with Cu/Mn diatomic pairs (Figure 8a) [29]. This nanozyme not only possesses excellent CAT-like and OXD-like activities but also serves as a porous nanocarrier with a near-infrared response and a photothermal conversion efficiency (η) of 42.8% at 1064 nm. Both experimental and theoretical studies have shown that the excellent catalytic activity of the nanozymes originated from the super-exchange interaction (SEI) between Cu and Mn atoms, which produced Cu^2+^/Mn^3+^ diatomic pairs with optimized electronic structures and significant synergistic effects, accelerating the rate-determining step of the cascade reaction. The porosity of the nanozyme enables it to have a high loading rate (~53.3%) for DOX. Under the weakly acidic conditions of the tumor, the nanozyme was able to convert hydrogen peroxide into singlet oxygen (^1^O_2_), inhibit the activity of heat shock proteins and P-glycoprotein (P-gp), improve the efficacy of photothermal therapy, and reduce the drug resistance of tumor cells. CuMn-DANs exhibited a higher relative ^1^O_2_ yield (43.4%) compared to CuCu-DANs (18.2%) and MnMn-DANs (5.7%). Both in vivo and in vitro tests demonstrated that the nanozymes achieved significant synergistic therapeutic effects.

The osteosarcoma malignancy damages a large percentage of the population, especially adolescents and young adults [103]. The development of osteosarcoma can lead to pain, fractures, and tumor metastasis. Osteosarcoma treatment requires effective anti-tumor therapy and advanced osteogenic techniques. At the same time, the accompanying antimicrobial properties are expected to ensure a good prognosis for osteosarcoma treatment in order to avoid frequent surgeries during bone repair and possible infections due to chronic osteomyelitis [104]. Wang et al. integrated highly active single-atom iron nanozymes (Fe SAN) into 3D-printed bioactive glass (BG) scaffolds for osteosarcoma treatment, bacterial killing, and subsequent osteogenesis (Figure 8b) [105]. The excellent Fenton-like catalytic activity and remarkable PTT effect of Fe SAC effectively eliminated osteosarcoma cells, exerting significant antibacterial and anti-osteomyelitis activities. The Fe SAN-BG scaffolds implanted into the bone defect site may accelerate bone marrow mesenchymal stem cells, accelerate osteoconduction and osteoinduction. This work broadens the application of SANs in integrated biomedical tissue engineering.

**Figure 8 ijms-24-15712-f008:**
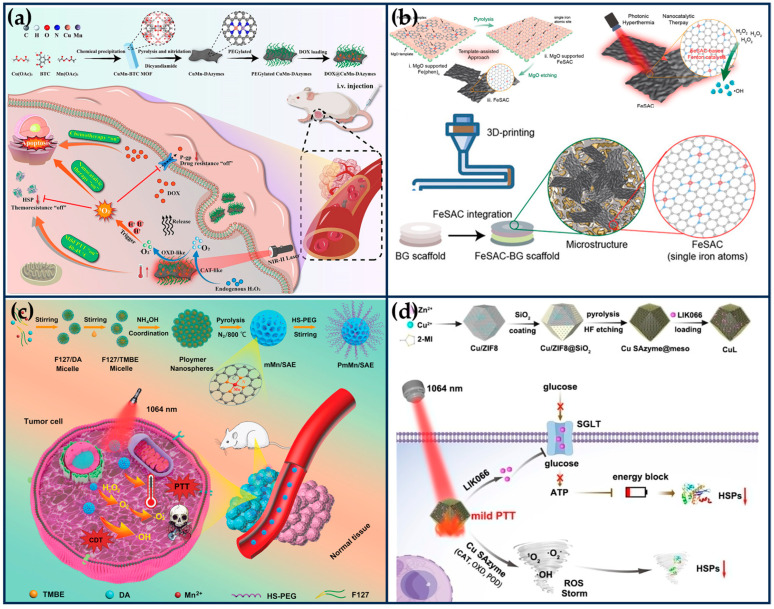
Synergistic tumor therapy with SANs combined with PTT. (**a**) Schematic illustration of the preparation of the DOX@CuMn-DANs, together with their main mode of action for synergistic ^1^O_2_-based nanocatalytic/mild photothermal/chemo-therapy (reproduced from [29] with permission from Elsevier). (**b**) Schematic illustration of the Fe SAN preparation process and integration of highly active Fe SAN into 3D printed bioactive glass (BG) scaffolds (reproduced from [105] with permission from John Wiley and Sons). (**c**) Schematic illustration for the preparation of Mn-based SANs and the application mechanism of the synergetic PTT and catalytic therapy (reproduced from [106] with permission from John Wiley and Sons). (**d**) Schematic illustration of the preparation process of CuL and the therapeutic strategy for mild PTT (reproduced from [107] with permission from John Wiley and Sons).

The NIR-II laser (1064 nm) has deeper tissue penetration and safety than NIR-I (808 nm) and produces thermal effects that can be synergized with other therapeutic modalities to eliminate tumors. Ye et al. successfully prepared PEGylated mesoporous manganese-based SANs (PmMn/SAN) using polydopamine via a coordination-assisted polymerization strategy (Figure 8c) [106]. PmMn/SAE exhibits excellent multi-enzymatic properties, including CAT-like, OXD-like, and POD-like activities, and not only catalyzes the conversion of endogenous H_2_O_2_ into O_2_ to alleviate the intra-tumoral hypoxia, but also transfers electrons to O_2_ to produce $O_{2}^{•-}$. Meanwhile, PmMn/SAE also induces apoptosis in cancer cells by generating highly toxic •OH through a Fenton-like reaction. PmMn/SAN also has excellent adhesion, biocompatibility, and photothermal conversion properties (η = 22.1%), and under laser irradiation at 1064 nm, the multiple ROS and photothermal properties synergized for better tumor inhibition. DFT calculations revealed the POD-like catalytic mechanism of PmMn/SAN, demonstrating the catalytic advantage of Pm Mn/SAE over H_2_O_2_. The in vitro and in vivo results demonstrated that PmMn/SAN could effectively kill cancer cells by photothermally enhanced catalytic therapy.

Excessive heat in PTT will inevitably damage normal tissues near the tumor, whereas the damage to cancer cells caused by mild thermotherapy is effortlessly repaired by stress-induced heat shock proteins (HSPs) [108]. The Pd SANs prepared by Chang et al. showed excellent POD-like and GSHOx-like activities and photothermal conversion properties [109]. Under the therapeutic environments of TEM and 42 °C, Pd SANs could produce •OH and consume GSH in large quantities, which induced lethal LPO, inhibited the expression of HSPs, and achieved complete tumor regression. To inhibit the protective function of HSPs more effectively, Chang et al. designed porous Cu SANs loaded with liogliflozin (LIK066), which can inhibit the glucose uptake of tumor cells by closing the sodium-dependent glucose transporter protein “valve” and effectively block the energy source for the synthesis of adenosine triphosphate (ATP), thereby inhibiting the synthesis of HSPs (Figure 8d) [107]. Concurrently, the ROS storm induced by Cu SAN could destroy HSPs in cancer cells. Through the two-pronged strategy, the LIK066-loaded Cu SAN showed efficient inhibition of HSPs and achieved a mild PTT. Specifically, the porous structure of ZIF8 was loaded with Cu ions and then covered with a mesoporous SiO_2_ layer on the surface of ZIF-8, resulting in the formation of Cu/ZIF8@SiO_2._ Cu/ZIF8@SiO_2_ was calcined under an argon atmosphere, and then the SiO_2_ shell layer was removed by hydrofluoric acid (HF) etching to obtain Cu SAN. Compared with the Pd SAN reported by their group [109], Cu SAN exhibited stronger cascade catalytic activities (CAT-like, OXD-like, and POD-like activities) and was capable of producing more ROS.

### 4.2. Sonodynamic Therapy (SDT)

Improving the intrinsic efficiency of sonosensitizers is an effective strategy to enhance the efficacy of SDT. Graphite-like phase carbon nitride (C_3_N_4_) is a semiconductor photocatalyst consisting of C and N elements, which has a high electron (e^−^)-hole (h^+^) separation efficiency and ROS generation ability under UV–visible irradiation. However, its application in the field of tumor SDT therapy is still limited by the drawbacks of poor water solubility, large particle size, low conductivity, and fast e^−^ and h^+^ pair complexation [110].

Previous work has shown that metal element doping (e.g., Fe, Cu, Mn) can narrow the bandgap of C_3_N_4_ and facilitate the separation of e^−^ and h^+^ pairs, thereby improving ROS generation efficiency [111]. Based on this, Feng et al. synthesized single-atom Fe-doped C_3_N_4_ semiconductor nanosheets (Fe-C_3_N_4_ NSs), which acted as a chemical-responsive sonosensitizer, effectively separating the e^−^ and h^+^ pairs and achieving a large amount of ROS generation at the melanoma site under ultrasound irradiation (Figure 9a) [112]. The N-coordinated holes and high-density homogeneous “six-fold cavities” of the C3N4 can effectively trap Fe ions. The doping of single Fe atoms can significantly improve the separation efficiency of e^−^-h^+^ pairs and exerts its high-efficiency POD-like activity to catalyze the generation of a large number of •OH, synergistically enhancing the SDT-mediated therapeutic effect. DFT studies and experimental results show that the doping of Fe atoms induces charge redistribution in C_3_N_4_ NSs, which enhances their synergistic SDT/chemokinetic activity, thus showing significant anti-tumor effects, thereby showing significant anti-tumor effects. This work extends the application of semiconductor-based inorganic acoustic sensitizers in tumor therapy.

### 4.3. Chemotherapy (CT)

Motivated to enhance the efficacy of SANs CDT, Zhu et al. constructed hollow carbon nanospheres loaded with oleanolic acid (OA) single-atom Fe-anchored (OA@Fe-SAN) by a template-mediated carbonization strategy [24]. The erythrocyte membrane was further loaded on its surface (OA@Fe-SAN@EM), which both prevented the leakage of OA and enhanced the accumulation of OA@Fe-SAN@EM in the tumor. Firstly, SiO_2_ nanoparticles were used as a template to introduce iron acetylacetonate during in situ polymerization of dopamine on its surface. Subsequently, it was pyrolyzed at 900 °C in a nitrogen atmosphere to anchor isolated Fe atoms on the N-doped C shell transformed from the PDA layer. Hollow N-doped carbon nanospheres supported by Fe SAN were obtained by etching SiO_2_ core with 4% hydrofluoric acid. Finally, OA was loaded into the hollow nanostructured Fe-SAC, and EM was coated on the surface of the hollow nanostructure. Excessive H_2_O_2_ in cancer cells could pass through the EM layer and produce * OH catalyzed by Fe-SAN, leading to EM breakage and release of OA. OA could effectively increase the expression of endogenous acyl-coenzyme A synthetase long-chain family member 4 (ACSL4) and enrich ROS-sensitive polyunsaturated fatty acids (PUFAs) to the cellular membranes, thus synergistically enhancing the CDT effect and exacerbating LPO by increasing the unsaturation of the cell membrane. In vivo experiments demonstrated that OA@Fe-SAN@EM NPs were able to inhibit tumor growth without significant side effects significantly. The strategy of enhancing CDT by adjusting membrane unsaturation proposed in this study opens up new ideas for enhancing the anti-tumor effect of SAN [24].

Zhang et al. synthesized a nanotherapeutic platform with tumor-targeting function by loading porous Fe SAN with DOX and encapsulating it with A549 CM, which enhances the tumor cell accumulation capacity of the therapeutic agent in the blood circulation time and combines the catalytic treatment with Fe SAN and the chemotherapy with DOX to improve the tumor inhibition efficiency.

### 4.4. Sonothermal Therapy (STT)

Ultrasound therapy allows ultrasound to reach deeper tissues (>10 cm) than NIR light (penetration ≈ 1 cm). SDT utilizes sonosensitizers to produce biotoxic ROS (e.g., ^1^O_2_ and •OH) to kill tumor cells [116]. Sonosensitizers (e.g., hematoporphyrins, photopigments, TiO_2_, fullerenes) require continuous, high levels of O_2_ to produce ROS, and hypoxia and antioxidant TME can limit the efficacy of SDT. High-intensity focused ultrasound is a non-invasive ultrasound therapy technique that uses non-ionizing ultrasound waves to heat or destroy solid tumors, which can increase the temperature of the tissue at the focal point to between 65~85 °C in a short period but can also cause non-specific thermal damage to normal tissue [117].

Electron-rich SANs offer the possibility of STT under low-intensity ultrasound irradiation. Qi et al. synthesized boron imidazole skeleton-derived nanocubes anchored with a single atom of Cu by a “B-H” interfacial domain-limiting ligand strategy (Figure 9b) [113]. The Cu SANs exhibit excellent sonothermal conversion properties as a result of strong intermolecular lattice vibrations when irradiated with low-intensity ultrasound. Moreover, Cu SANs were able to exhibit strong biocatalytic activity, catalyzing the generation of large amounts of toxic •OH. In vitro and in vivo evaluations confirmed that the sonothermal–catalytic synergistic therapeutic strategy mediated by Cu SANs effectively inhibited tumor proliferation (tumor suppression rate of 86.9%) and enhanced the survival rate (100%) of the MDAMB-231 tumor-bearing nude mice after a single injection and irradiation. These findings provide a novel approach to designing multifunctional nanoplatforms for precision cancer therapy.

### 4.5. Gas Therapy (GT)

Recently, the strategy of gas cancer therapy has received extensive attention. Some gas molecules, such as nitric oxide (NO), carbon monoxide (CO), hydrogen (H_2_), and sulfur dioxide (SO_2_), can be used to induce cancer cell death [118,119,120,121]. Gas therapy has more substantial penetration and retention effects and better diffusion in tumor tissues than radiotherapy, chemotherapy, and other therapies.

In order to enhance the targeting and the efficacy of SAN therapy, Chen et al. developed a cell membrane-encapsulated SANs targeted therapy system (Figure 9c) [114]. Firstly, the pyrolysis process prepared Cu SAN using polydopamine nanoparticles as a carbon source with a 0.9% mass fraction of Cu atoms, exhibiting high POD-like activity under weak acid conditions. Next, the targeted composite system was prepared by extruding platelet vesicles (PV), O_2_ prodrug (benzothiazole sulfinic acid, BTS), and Cu SAN together through an extruder. Among them, p-selectin on the platelet membrane can specifically recognize CD44 receptors on tumor cells to achieve tumor targeting. The sulfite structure of water-soluble BTS is easily hydrolyzed to SO_2_ in the weakly acidic environment of tumors. SO_2_ further reacts with H_2_O_2_ to generate toxic •SO_3_, which increases the expression of pro-apoptotic proteins calpain I, Bax, and p53 and inhibits the expression of anti-apoptotic protein bcl-2, thus inducing apoptosis. At the same time, it will consume excessive GSH in tumor cells, destroy the intracellular redox balance, and reduce the ROS tolerance of tumor cells, which further enhances the nano-catalytic therapeutic effect of Cu SAN. These properties of the composite system were verified in a mouse model of MFC peritoneal metastases, where the composite system inhibited 90% of the tumors and showed good biocompatibility.

### 4.6. Magnetic Resonance Imaging (MRI)

Among various biomedical imaging modalities, MRI is an effective and noninvasive diagnostic method for disease evaluation and diagnosis, especially for early detection and precise localization of tumors [122,123]. Due to the low sensitivity of MRI, scientists have utilized contrast agents (e.g., Gd-chelators) to enhance the lining of clinical imaging by increasing the spin relaxation rate of water molecules in vivo [124]. With the rapid development of nanotechnology, Gd-based nanomaterials exhibit better relaxation properties than Gd-chelates by increasing the rotational correlation time (τr) of geometrically confined Gd elements, and they can be enriched in tumors [125]. However, the inability of the Gd atoms inside the nanoparticles to contact water molecules leads to the waste of Gd atoms and increases the medical cost [126]. Moreover, the release of toxic Gd^3+^ has the risk of causing nephrotoxicity [127]. It is urgent to develop a Gd-based nanocontrast agent with biocompatibility, tumor targeting capability, stability, and a high relaxation rate for T1-weighted tumor MRI. Single-atom Gd nanomaterials can mimic the coordination environment of paramagnetic metals and ligands at the atomic level and maximize the exposure of Gd atoms while providing better structural stability, which is conducive to enhanced imaging contrast.

Therefore, Liu et al. prepared GaSA by loading single-atom Gd onto hollow N-doped carbon nanorods via a template-mediated nitridation strategy (Figure 9d) [115]. Subsequently, its surface was modified with DSPE-PEG_2000_-NH_2_ to enhance its dispersibility and biocompatibility. Ga atoms are coordinated by six N atoms and two O atoms and thus exhibit higher stability than gadolinium diethylenetriamine pentaacetic acid (Gd-DTPA) and Gd_2_O_3_ nanoparticles, as well as minimal hematotoxicity, nephrotoxicity, and hepatotoxicity. Ga-SA has a longitudinal relaxation rate (r1) value of up to 11.05 mM^−1^ s^−1^ and exhibits a more pronounced magnetic resonance (MR) contrast enhancement on tumor tissues at 24 h after administration, which is due to the exchange of relaxed water molecules being faster in Gd-SA than in Gd-DTPA.

Regulating the morphology of the Gd-SA material increases the hydrophilicity of the material. It reduces the resistance of water molecules to diffusion, which is favorable for further improving the relaxation rate in MRI. Luo et al. obtained GdSA nanomaterials with a bowl-like structure (feGd-NxC) by controlling the etching process [128]. The bowl-shaped structure increased the hydrophilicity of the matrix, and water molecules were more accessible to the Gd^3+^ center, enabling the r1 value of feGd-NxC to reach 34.02 mM^−1^ s^−1^ (3T). In addition, Gd-DTPA was only used for T1-weighted MRI, and feGd-NxC has the ability of T1/T2 dual-mode MRI, which can effectively avoid false-positive signals. These works provide a new direction for designing safe and effective nanoprobes for MRI-based disease diagnosis.

### 4.7. Immunotherapy (IT)

Despite tremendous advances in cancer treatment, it remains one of the leading causes of death today. Cancer IT is recognized as one of the most promising ways to completely cure cancer by triggering a systemic and sustained immune response to promote tumor regression and inhibit tumor metastasis [129,130,131,132,133]. However, tumor cells often exhibit low immunogenicity to evade recognition by immune cells. Currently, one of the main anti-tumor strategies is to trigger apoptosis by anticancer drugs. The clearance of apoptotic tumor cells by macrophages promotes the immunosuppression of the tumor microenvironment (TME) and attenuates the stimulation of the host immune system. Therefore, in order to activate anti-tumor immune activity, other non-apoptotic cell death pathways need to be activated to enhance tumor immunogenicity. Ferroptosis is a classic non-apoptotic cell death pathway characterized by iron accumulation and lipid peroxidation (LPO) [134]. LPO associated with ferroptosis can act as a “find me” signal to promote recognition and phagocytosis of tumor antigens by dendritic cells, thereby activating cytotoxic T lymphocytes and enhancing tumor IT [135].

Although ROS can promote ferroptosis, intracellular overexpression of reduced glutathione (GSH) in tumor cells eliminates ROS bursts at the tumor site, making it difficult to activate ferroptosis with most current ROS-based therapeutic strategies, including PDT, CDT, and SDT. Therefore, consuming endogenous GSH while increasing ROS at the tumor site would effectively activate initial tumor immunogenic ferroptosis. Nanozymes can specifically react with excessive hydrogen peroxide (H_2_O_2_) and GSH under unique TME and play an essential role in ROS-mediated cancer treatment [136].

The formation of catalytic reaction intermediates with high energy barriers severely limits the catalytic activity of SANs due to the excessive binding strength between the transition metal monoatomic sites and the electron-donating intermediates. Compared with SANs, non-homogeneous dual-atom site catalysts can utilize two different adjacent metal atoms to achieve their functional complementarity and synergy. In particular, the energy barrier of the reaction intermediates can be modulated by the electronic interactions between the two neighboring metals [137]. Therefore, preparing non-heterogeneous diatomic nanozymes with ROS generation and GSH removal capabilities is important for ferroptosis-based antitumor IT.

Liu et al. obtained N-doped graphene-encapsulated FeCo nanocages by pyrolysis under N_2_ with FeCo Prussian blue analog (FeCo PBA) as a prerequisite, followed by acid etching to obtain FeCo/Fe-Co DANs with OXD-like, GSHOx-like, CAT-like and POD-like activities (Figure 10a) [138]. Due to the porosity of carbon nanomaterials, the FeCo/Fe-Co DAN was further loaded with phospholipase A2 (PLA2) and lipoxygenase (LOX) with loading rates of 57.3 and 54.9%, respectively. This composite nanozyme induced initial immunogenic tumor iron death through its own multi-enzyme mimetic activity and upregulated AA expression and synergistically induced ACSL4-mediated immunogenic tumor ferroptosis in conjunction with CD8^+^ T cell-derived IFN-γ. During the treatment, FeCo/Fe-Co DAN/PL could decompose overexpressed H_2_O_2_ in tumor tissues to O_2_, which further reacted with free AA released from PLA2-catalyzed phospholipids to generate AA-OOH under LOX catalysis, and AA-OOH can generate a large amount of ferroptosis (^1^O_2_) through the Russell mechanism. Meanwhile, FeCo/Fe-Co DAN can catalyze the production of $O_{2}^{•-}$ and •OH from O_2_ and H_2_O_2_, which can not only exacerbate the oxidative damage to tumor cells but also cooperate with LOX to promote LPO. Compared with Fe or Co SANs, the synergistic interaction between adjacent atoms of Fe-Co DAN greatly facilitated its induction of ROS generation. In addition, FeCo/Fe-Co DAN mimics GSHOx, oxidizes GSH to GSSG, reduces the antioxidant capacity of the tumor, effectively inhibits GPX4 activity, and promotes ROS- and LOX-induced LPO. The above effects cause initial immunogenic ferroptosis, further activating the systemic antitumor immune response. Most importantly, CD8^+^ T cell-derived IFN-γ can reprogram ACSL4-associated phospholipids by binding to AA, further triggering efficient cascading immunogenic tumor ferroptosis. In conclusion, this six-enzyme co-expressed composite nanozyme can synergistically promote irreversible cascade immunogenic tumor ferroptosis by synergizing multiple pathways, including ROS storm, GSH/GPX4 depletion, LOX catalysis, and IFN-γ-mediated ACSL4 activation, which is highly feasible for future oncology applications [138].

IT treats cancer by stimulating a non-specific immune response in the body but still causes some adverse effects associated with systemic treatment. He et al. prepared phenolic Pd SANs (DA-CQD@Pd) by in situ synthesis of Pd single atoms on catechol-grafted carbon quantum dots [139]. Bioadhesive injectable hydrogels consisting of DA-CQD@Pd SAN and the immune adjuvant CpGODN were formed by SAN-catalyzed radical polymerization for local immunomodulation and catalytic enhancement of IT. DA-CQD@Pd SAN has a dual catalytic mechanism, whereby the Pd single atom and the catechol–quinone redox pair on the DA-CQD can catalyze in parallel the H_2_O_2_/APS to produce •OH, which induces ICD in tumor cells and generates tumor-associated antigens in tumor lysates, triggering an anti-tumor immune response. The slow release of the immune adjuvant CpGODN further enhances the anti-tumor immune response. By combining with the immune checkpoint inhibitor anti-PD-L1, the hydrogel maximized local immunomodulation and prevented tumor recurrence and metastasis.

**Figure 10 ijms-24-15712-f010:**
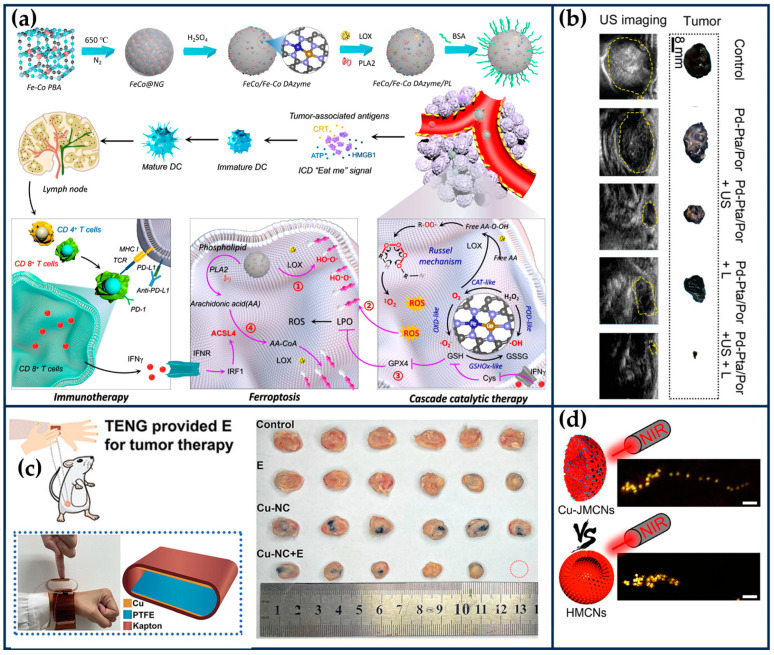
(**a**) Schematic illustration of the synthetic process and therapeutic mechanism of FeCo/Fe–Co DAzyme/PL (reproduced from [138] with permission from the American Chemical Society). (**b**) Tumor images and ultrasound imaging (reproduced from [140] with permission from John Wiley and Sons). (**c**) Optical photograph of the driving TENG worn on humans and its structural schematic diagram and optical photograph of the tumors on day 14 (reproduced from [141] with permission from John Wiley and Sons). (**d**) Motion trajectory comparison of Cu-JMCNs and HMCNs in the presence of NIR light irradiation (reproduced from [142] with permission from the American Chemical Society).

### 4.8. Multiple Modality Therapy

Du et al. designed novel and facile Pd-single-atom coordinated porphyrin-based polymer networks as biocatalysts for chemo/acoustic/photo-trimodal tumor therapy [140]. Atomic morphology and chemical structure analyses demonstrated that the biocatalyst consisted of an atomic-level Pd-N coordination network, and Pd-N_2_-Cl_2_ was verified as the catalytic center by XAS. POD-like catalytic activity measurements showed that Pd-Pta/Por can generate abundant •OH for CDT. Ultrasound irradiation or laser excitation could enhance the ability of porphyrin-like acoustic/photosensitizer in Pd-Pta/Por to produce ^1^O_2_. DFT results demonstrated that Pd-Pta/Por could completely convert H_2_O_2_ into two •OH, which ensured the efficient generation of ROS. The in vitro and in vivo results also verified the chemical/acoustic/photo-trimodal synergistic oncological efficacy of Pd-Pta/Por (Figure 10b).

Zhu et al. encapsulated Pd-C SANs and camptothecin in agarose hydrogel to synergistically enhance the anti-tumor activity by combining PTT, CT, and catalytic therapies [96]. The Pd-C SAN was able to convert near-infrared (NIR) laser light into heat, which led to the degradation of the hydrogel and the slow release of camptothecin. The camptothecin not only killed the tumor cells but also activated nicotinamide adenine dinucleotide phosphate oxidase, increased the H_2_O_2_ content in the tumor, further enhanced the POD-like activity of SANs, and promoted the immunogenic death of the tumor.

### 4.9. Other Therapeutic Approaches

Non-invasive exogenous stimuli, such as light, electric fields, ultrasound, and magnetic fields, have enhanced the catalytic activity of nanozymes [143]. Among them, the wide and deep driving field of the electric field can cover the tumor tissue well, kill the cancer cells directly, activate the anti-tumor immunity of the system, or trigger the catalytic activity of the nanomaterials [144]. Strong electric fields and high voltages from conventional power supply can have adverse effects on the human body, so Zhong et al. developed a self-powered and wearable triboelectric nanogenerator (TENG) to generate a milder and safer electric field (Figure 10c) [141]. The TENG can convert mechanical energy from daily human activities into electrical impulses based on contact charging or electrostatic induction. Moreover, they found that the electric field emitted by the self-supplied TENG could elevate the POD-like, CAT-like, OXD-like, and GSHOx activities of Cu-NC, which promotes the production of free radicals and amplifies the oxidative damage and death of cancer cells. Computational studies show that electric field stimulation promotes H_2_O_2_ adsorption on the CuN_4_ active site, increases Cu dxy orbitals near the Fermi energy level, and shifts the d-band center of Cu. This work opens up new perspectives for improving SANs for cancer therapy.

Current SANs mainly rely on passive transport to the tumor site. The lack of active delivery capability can lead to limited depth of tumor penetration and reduce the efficacy of SANs. Nanomotors can convert a variety of energies into autonomous motions. Limited by the ionic strength in the human body, the two main mechanisms by which nanomotors are effectively self-driven in bioliquids are self-diffusion and self-thermophoresis, enabling them to target tumors actively, enhance cancer cell membrane adhesion and increase tumor penetration depth [145,146]. Therefore, by rationally designing the morphology of SANs and using the photothermal properties of SANs, SANs can be constructed into self-thermally driven nanomotors, combined with the high catalytic activity of ROS of SANs, to improve the tumor treatment effect of SANs further.

Xing et al. constructed a single-atom copper-coordinated N-doped jellyfish-like mesoporous carbon nanomotor (Cu-JMCNs) through an emulsion-induced interfacial anisotropic assembly strategy [142]. The mesoporous surface gives the prepared nanomotors good dispersion and stability in an aqueous solution. Cu single atoms can catalyze the H_2_O_2_ to generate toxic •OH for CDT. Under the irradiation of near-infrared light, the jellyfish-like structure of Cu-JMCNs can generate asymmetric thermal distribution to achieve more effective self-thermophoresis directional propulsion than hollow mesoporous carbon nanospheres (HMCNs) (Figure 10d). The anti-tumor enhancing effect of Cu -JMCNs on different cancer models was systematically explored. For 2D cancer cells, NIR light propulsion only slightly increased the CDT effect by 8.4%, possibly due to the limited intracellular H_2_O_2_ concentration. On the contrary, for 3D MTSs and in vivo tumors, the improved cell membrane adhesion and cellular uptake by NIR light propulsion could significantly increase the penetration depth of Cu-JMCNs, effectively enhancing the single-atom CDT effect (38.6% and 22%, respectively). This work provides a rational design and preparation strategy for constructing self-driven SANs, which offers a new path for active nanomedicine development.

Zhu et al. encapsulated the proton pump inhibitor (PPI) and Cu SAN in PV. PPI inhibits the activity of vesicular proton pumps (V-ATPases), which inhibits ATP hydrolysis and promotes the accumulation of acid in the cell, and also affects the metabolism of Gln and reduces the content of intracellular GSH, which enhances the catalytic therapeutic effect of Cu SAN [147].

## 5. Application of SANs in Tumor Diagnostics

Nitric oxide (NO) is an endogenously produced signaling molecule that acts intracellularly and intercellularly [148]. NO is a natural biomolecule with redox activity, produced by the oxidation of L-arginine catalyzed by nitric oxide synthase (NOS) [149]. The controlled release of NO significantly impacts the maintenance of vascular homeostasis. In response to mechanical forces (circumferential stretch or fluid shear stress), endothelial cells produce excessive amounts of NO, triggering a cascade of biological reactions that can dysregulate oxidative homeostasis and lead to diseases such as neurodegenerative disorders, autoimmune processes, and cancer [150]. For example, NO in the brain, in the threshold concentration range (nM) regulates synaptic transmission and neuronal activity. However, if the NO concentration exceeds the threshold range (up to μM), it will induce oxidative stress and cause neuronal damage [151]. Therefore, developing real-time sensing platforms for NO under normal and pathological conditions is crucial for human health detection. Monitoring intracellular NO levels requires sensors with sufficient sensitivity, transient recording capability, and biocompatibility. Zhou et al. designed a Ni single-atom-based chemical sensor for NO detection in the live cell environment [152]. Ni single atoms anchored on N-doped hollow carbon spheres served as excellent catalysts for the electrochemical oxidation of NO. DFT calculations showed that the Gibbs free energy required for NO activation by Ni SACs/N-C was drastically reduced, thus exhibiting superior performance to Ni-based nanomaterials. Furthermore, Ni SACs-based flexible, stretchable sensors with high biocompatibility and low nanomolar sensitivity enable real-time monitoring of NO content in cells under drug and stretch stimulation. This study provides a paradigm for designing and developing SACs-based sensing devices for potential medical monitoring.

By modulating NO concentration in the in vivo microenvironment and utilizing its vasodilatory function, NO can be used for wound healing and anti-tumor therapy of vascular regulation. Therefore, accurate determination of NO concentration is of great significance for understanding its function and various life activities of organisms. Hu et al. used Co SAN to construct active materials for NO electrochemical sensors (Figure 11a) [153]. The constructed flexible sensor can be connected to portable electronic devices for the in situ processing of signals and wirelessly communicate with the user interface via Bluetooth, enabling highly sensitive in situ monitoring of NO at the cellular and organ levels.

SANs on MXenes show promising applications in cancer diagnostics. By calcination, single Au atoms are embedded into the Ti vacancies of MXene, redistributing the charge of MXene through metal–carrier interactions [154]. At the SA Au sites, H_2_O_2_ gains electrons and is converted into •OH, leading to a significant increase in electrochemiluminescence intensity, which is twice that of AuNPs-MXene. In addition, during the synthesis process, part of the Ti is oxidized to TiO_2_, constructing a heterojunction of MXene and TiO_2_, accelerating free radical generation, and facilitating the electrochemical signals. For the clinical detection of miRNA-187 in triple-negative breast cancer tumor tissues, the luminescence intensity was positively correlated with the concentration of miRNA-187.

MoSA-N_3_-C was successfully applied for selective and sensitive analysis of xanthine in human urine samples, contributing to the early diagnosis of renal lesions [82].

Photoelectrochemical immunoassay is a novel detection platform based on changes in photocurrent induced by photosensitive nanomaterials recognizing antigens and antibodies. Due to their excellent catalytic activity and selectivity, SACs have also been used to construct novel photoelectrochemical sensor platforms. Zeng et al. synthesized single-atom platinum-supported hollow cadmium sulfide and constructed a photoelectrochemical biosensor using it as an etched substrate for detecting exosomes [155]. Qin et al. utilized single-atom platinum-supported CdS nanorods with ultra-high photoactivity to establish an advanced photoelectrochemical sensor for monitoring disease-related biomarkers [156]. However, single-atom-loaded pure CdS also suffers from the drawbacks of unstable photocorrosion and rapid compounding of visible light-generated photogenerated charges in the presence of light. To overcome these problems, Li et al. prepared single-atom platinum-anchored Zn_0.5_Cd_0.5_S nanocrystals as a novel photoelectrochemical immunosensor for photocurrent determination of prostate-specific antigens (PSA) (Figure 11b) [157]. The doping of Zn increases the minimum energy of the conduction band and the mobility of photogenerated holes/electrons of the CdS crystals, which is favorable for the reduction of bulk hole–electron complexation and photogenerated hole oxidation of divalent sulfur ions. The introduction of single-atom Pt can likewise enhance the photocurrent and further improve the sensitivity of the immunoassay. When the target PSA is detected under acidic conditions, CuO carried in the sandwich immunoreaction releases Cu^2+^, which undergoes an ion-exchange reaction with single-atom Pt-anchored Zn_0.5_Cd_0.5_S to form a weakly photoreactive substance, CuxS, and decrease the photocurrent density. The photoelectrochemical immunosensor based on single-atom platinum-anchored Zn_0.5_Cd_0.5_S could detect PSA in the 1.0–10,000 pg/mL range, with a detection limit of 0.22 pg/mL. This photoelectrochemical immunosensor also has the advantages of superior reproducibility, high precision, and selectivity, which can provide an important idea for the early screening and diagnosis of tumors.

**Figure 11 ijms-24-15712-f011:**
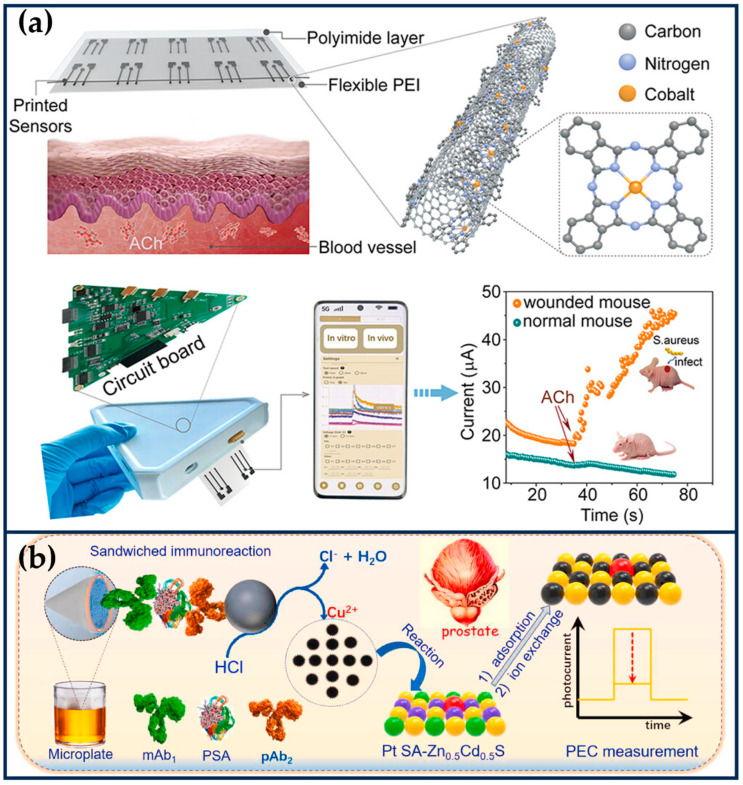
(**a**) Schematics of the integrated electrochemical NO-sensing system (reproduced from [153] with permission from the American Chemical Society). (**b**) Ion-exchange reaction between single-atom platinum-anchored Zn_0.5_Cd_0.5_S and the released Cu^2+^ from CuO nano label (reproduced from [157] with permission from Elsevier).

## 6. Concluding Remarks and Future Perspectives

In recent years, numerous engineered SANs have been developed for treating tumors and have shown promising therapeutic effects in in vitro and in vivo studies. The high metal utilization and enzyme-like catalytic activity of SANs allow them to maximize therapeutic efficacy with minimal ionic toxicity, and their well-defined electronic geometries allow scientists to investigate their catalytic activity and mechanism both experimentally and by theoretical calculations. Here, we summarize a representative selection of SANs, showing their active centers, preparation methods, metal loading, and therapeutic approaches to tumor treatment (Table 1). The relevant kinetic parameters of these SANs are summarized in Table 2. The primary purpose of this review is to summarize what kinds of SANs are used in oncology and what approaches are taken to improve the efficiency of oncology. This will provide a better understanding of the current development of SANs and provide exceptional insights for designing better oncology platforms. Despite the great success of SANs, there are still some issues that need to be challenged:

(1) The loading of single metal atoms is also difficult to control effectively, primarily attributed to the Gibbs–Thomson effect, which often leads to agglomeration of the anchored single atoms. Although nitrogen doping can alleviate the aggregation, simply increasing the nitrogen content cannot always improve the single atoms loading. Exploiting approaches to address the low loading of metal atoms is also a pressing issue for the future.

(2) Xin et al. synthesized a series of single-atom catalysts of single metal elements on nitrogen-doped carbon materials by high-temperature carbonization [158]. However, a library of single-atom catalysts supported on other materials has not yet been established. It is necessary to understand the formation mechanism of SAC comprehensively, explore the influence of metal oxidation state, coordination number, bond length, coordination element, and loading amount on the activity, reveal the inherent law between them, guide the preparation of single-atom catalysts, and provide assistance for the application of SAC in biomedicine.

(3) Recently, nanozymes have demonstrated the potential to enhance in vitro and in vivo anti-tumor effects by catalyzing the conversion of metabolic small molecules, such as glucose, glutathione, NADH, and H_2_O_2_. However, designing single-atom nanozymes for catalyzing the transformation of physiological molecules, especially for crucial signaling metabolites in tumor development, is still very challenging. This is one of the difficulties that need to be overcome for further application of SANs in cancer therapy in the future. For example, tumor lactate is an essential physiological signaling molecule that plays a role in tumor malignant progression, angiogenesis, immunosuppression, and therapy resistance [159]. The rational design of single-atom nanozymes with lactate oxidase (LOX) mimetic activity could facilitate lactate-responsive tumor therapy. However, oxidation of the α-C-sp(3)-H bond of lactate to pyruvate under mild conditions is exceptionally challenging for conventional chemical catalysts due to the high bond energy of the C-H bond [160].

(4) Preparation of more highly active SANs is required. COF materials’ high specific surface area, highly ordered periodic structure, high porosity, and ease of functionalization make them excellent carriers for the construction of nanozymes. For instance, Kumar et al. modified citrate-capped Au nanoparticles on PEI-COF nanosheets through electrostatic interactions [161]. Due to the abundant amino groups on the surface of PEI-COF nanosheets, the high specific surface area well, and the porosity, the contact between dissolved O_2_ and Au nanoparticles was increased, which improved their photocatalytic activity and was well applied in cancer cell detection, organic degradation, and antimicrobial applications. In addition, COF-based single-atom catalysts are also developing rapidly, and the loaded single atoms can increase the vacant d-atomic orbitals to modulate their electronic and catalytic properties and have been successfully applied in the fields of photocatalysis and electrocatalysis. Therefore, constructing novel COF-based SANs and applying them to tumor diagnosis and treatment is a future development direction. Some studies have shown that the presence of metal nanoclusters or nanoparticles in SANs positively affects the activity of SANs, which provides a direction for the preparation of highly efficient SANs. Many SACs with excellent redox capacity are well used in electrocatalysis and photocatalysis, so the biomedical value of these SACs with excellent performance can be explored.

(5) Most of the SANs are currently capable of treating a limited number of cancer types, and most of them remain at the stage of laboratory validation. Therefore, there is a need to validate the effectiveness of SANs against more types of tumors and to vigorously promote the marketing of SANs so that SANs can become anti-tumor drugs that truly benefit humankind.

**Table 1 ijms-24-15712-t001:** Recent examples of SANs for cancer therapy.

SANs	Active Center	Metal Loading	Enzyme-like Activity	Free Radicals and Their Detection	Combined Therapeutic Means	Tumor Model	Preparation Method	Ref.
DOX@CuMn-DAzymes	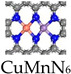	5.3 wt% Cu;4.1 wt% Mn	CAT-likeOXD-like	O2•− (BMPO)^1^O_2_ (TEMP)	ESR	CT; PTT	HeLa/ADR tumor bearing mice	MOF-mediated carbonization strategy	[29]
BSA-Cu SAN	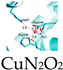	5.4 wt%	POD-like	•OH (DMPO)	ESR	BSA modification	HCT116 cells on BALB/c nude mice	Coordination reduction strategy	[41]
PV@BTS@Cu SAN	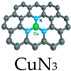	0.9 wt%	POD-like	------	------	GT; Proton pump inhibitors; CM modification	MCF tumor bearing mice;MGC803 tumors bearing BALB/c nude	High temperature carbonization strategy	[114]
FeCuNC@CM	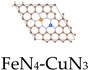	------	POD-like	•OH (isopropanol, DMPO)	UV, ESR	PTT; CDT; CM modification	4T1-Luc Subcutaneous model.	Template-assisted synthesis strategy	[28]
Gd-SA@DSPE-PEG 2000-NH_2_	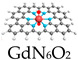	0.42wt %	------	------	------	DSPE-PEG 2000-NH_2_ modification	4T1-bearing BALB/c mice	Template-mediated carbonization strategy	[115]
feGd-NxC	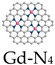	≈2.7 wt%	------	------	------	MRI	4T1 tumor bearing female BALB/c mouse	Template-mediated carbonization strategy	[128]
FeCo/Fe-Co DAzyme/PL@BSA	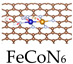	------	OXD-like; GSHOx-like; CAT-like; POD-like; LOX-like; PLA2-like	O2•− (DMPO)^1^O_2_ (TEMP)•OH (DMPO)	ESR	IT	4T1 single tumor-bearing mice model	Prussian blue analog-mediated carbonization strategy	[138]
Cu-JMCNs	Cu-N_4_	------	POD-like	•OH (disodium terephthalate)	FL	Nanomotor, PTT	MCF-7 cancer-bearing female Balb/c mice	Emulsion-induced interfacial anisotropic assembly strategy	[142]
MoOx-Cu-Cys-PVP@RGD	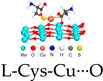	10.10 wt%	CAT-like; POD-like	•OH (DMPO)	ESR	Radiodynamic therapy; targeting αvβ3 integrin	4T1 tumor-bearing mice models	Coordination-driven self-assembly strategy	[22]
S-N/Ni PSAE	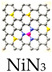	0.96 wt%	POD-like, GSHOx-like	•OH (DMPO)	ESR	------	4T1 tumor-bearing mice models	Anion exchange strategy	[23]
OA@Fe-SAC@EM NPs	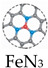	0.64 wt%	POD-like	•OH (DMPO)	ESR	CM modification	MCF-7 tumor-bearing nude mice	Template-mediated carbonization strategy	[24]
MC_2/3_Cp-SAE@Ce6@@PVP	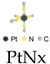	8.48 wt%	CAT-like	^1^O_2_ (SOSG)	FL	PDT	Balb/c mice with 4T1 tumor model	“missinglinker-confined coordination” strategy	[65]
IrN_5_ SAN@Cer@	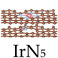	------	OXD-like; POD-like; CAT-like; NADHOD-like	O2•− (DMPO)•OH (DMPO)	ESR	------	4T1 tumor-bearing mice	MOF-mediated carbonization strategy	[25]
Cu/Zn DSAN@PEG	CuN_4_ZnN_4_	Cu 1.4 wt.%Zn 3.09 wt.%	POD-like	•OH	------	PTT	B16F10 tumor-bearing mouse model	MOF-mediated carbonization strategy	[54]
FeCo-DIA/NC@HA	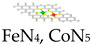	Fe 0.35 wt%, Co 1.32 wt%	POD-like;OXD-like	O2•− (DMPO)^1^O_2_ (TEMP)•OH (DMPO)	ESR	HA modification	HeLa tumor-bearing female BALB/c nude mice	MOF-mediated carbonization strategy	[27]
FeNC-edge	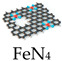	0.72 wt%	POD-like;OXD-like ;OXD-like	O2•− (DMPO)•OH (DMPO)	ESR	------	4T1 tumor-bearing mice model	H_2_O_2_-mediated generation of edge sites	[80]
Pt@IrSACs/RBC)	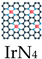	0.11 wt%	POD-like; GSHOx-like	•OH (DMPO)	ESR	RBCs modification	4T1 tumor-bearing mice	MOF-mediated carbonization strategy	[47]
Cu/TiO_2_@PEG	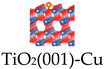	7.06 wt%	POD-like	^1^O_2_ (TEMP)•OH (BMPO)	ESR	SOD CDT	4T1 breast tumor-bearing mice	A reformative wrap–bake–strip method	[63]
GOD@FeN4-SAzyme	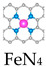	8.75 wt%	POD-like; GSHOx-like	------	------	GOD-assisted cascade catalysis; X-ray enhanced catalytic activity	4T1 tumor-bearing mice	Inorganic salt template method	[88]
Fe-CDs@Ang	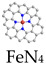	3.4 wt%	POD-like; OXD-like; SOD-like; CAT-like; TPx-like; GPx-like	O2•− (DMPO)•OH (DMPO)	ESR	BBB crossing; tumor targeting	Orthotopic U87MG GBM-bearing mouse model	Microwave-assisted pyrolysis strategy	[62]
(Fe, Pt) SA-N-C-FA-PEG	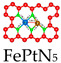	Fe 0.42 wt%;Pt 0.31 wt%	POD	•OH (BMPO)	ESR	------	4T1 tumor-bearing Balb/c mice	Secondary doping strategy	[26]
macDNAFe/PMCS SAzymes	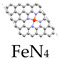	1.68 wt %	OXD, POD, GSH—consuming	O2•− (hydroethidine)^1^O_2_ (1,3-diphenylisobenzofuran)	FL	------	CT26 tumor-bearing mice and MCF-7 tumor-bearing mice	MOF-mediated carbonization strategy	[100]
DA-CQD@Pd	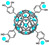	------	POD-like	•OH (DMPO)	ESR	------	CT26 Balb/C mice bearing tumors	Coordination reduction strategy	[139]
Pd-C SAzyme	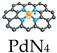		POD-like	•OH (DMPO)	ESR	CT; PTT	CT26 Balb/C mice bearing tumors	Template-mediated carbonization strategy	[96]
PmMn/SAE	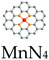	1.56 wt.%	POD-like; CAT-like; OXD-like	O2•− (DMPO)•OH (DMPO)	ESR	PTT	U14 tumor-bearing mice	Coordination-assisted polymerization self-assembly strategy	[106]
MoS_2_@SA-Fe	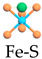	1.71 wt%	POD-like	•OH (DMPO)	ESR	------	T1 murine breast cancer model	One-pot solvothermal method	[77]
FeN_5_ SAzyme	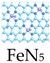	------	POD-like	•OH (terephthalic acid)	FL	------	4T1 tumor-bearing mice	Melamine-mediated two-step pyrolysis strategy	[81]
C-NFs@AFt@DOx	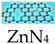	27.3 wt%	POD-like	•OH (DMPO)	ESR	CT	MCF-7/ADR tumor-bearing mice models	“ PDA-assisted morphology fragmentation “ strategy	[99]
PFCE@Co/A-TiO_2_ SAzyme	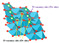	1.6 wt%	CAT-like; OXD-like	O2•− (DMPO)	ESR	MRI	4T1 xenografted BALB/c mice	Cation exchange strategy	[162]
Ru SAEs	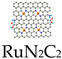	1.2 wt%.	POD-like; OXD-like; GSHOx-like	O2•− (DMPO)•OH (DMPO)	ESR	PTT	4T1 murine breast tumor model	Pyrolysis coordination complexation strategy	[70]
MoSA–Nx–C	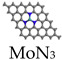	------	POD-like	------	------	Detection of xanthine in urine	------	MOF-mediated carbonization strategy	[82]
Pd SAzyme@PEG	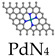	0.18 wt %	POD-like; GSHOx-like	•OH (DMPO)	ESR	PTT	4T1 xenograft BALB/c mice	MOF-mediated carbonization strategy	[109]
Pd-Pta/Por	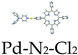	15.74 wt%	POD-like	•OH (DMPO)	ESR	SDTPDT	B16F10 tumor xenografts in Balb/c male mice	Coordination-driven self-assembly strategy	[140]
OxgeMCC-r SAE	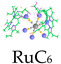	2.23 wt%	CAT-like	------	------	PDT; MRI	4T1 tumor bearing mice	Coordination-driven self-assembly strategy	[61]
Cu-HNCS	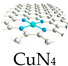	0.18 wt%,	POD-like;OXD-like	O2•− (BMPO)•OH (DMPO)	ESR	------	4T1 tumor-bearing mice	MOF-mediated carbonization strategy	[60]
HA-NC Cu	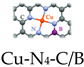	0.17 wt%	POD-like	•OH (DMPO)	ESR	STT	MDA-MB-231 Tumor-bearing nude mice	Confined coordination strategy at “B-H” interface	[113]
Fe-C_3_N_4_ NSs	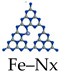	0.16 wt%	POD-like	^1^O_2_ (TEMP)•OH (DMPO)	ESR	SDT	B16F10 tumor-bearing mice	One-step pyrolysis strategy	[112]

Fluorescent (FL); Electron spin resonance (ESR); Ultraviolet (UV).

**Table 2 ijms-24-15712-t002:** The kinetic constants of SANs for cancer therapy.

Materials	Enzyme-like Activity	Substrate	Km (mM)	Vmax (M s^−1^)	kcat (s^−1^)	Ref
DOX@CuMn-DAzymes	OXD-like	TMB	0.59	8.32 × 10^−8^	------	[29]
	CAT-like	H_2_O_2_	17.8	3.23 × 10^−5^	------	
BSA-Cu SAN	POD-like	H_2_O_2_	0.48	17.09 × 10^−8^	3.65 × 10^−3^	[41]
FeCuNC	POD-like	H_2_O_2_	32.03	9.04 × 10^−7^	------	[28]
FeCo/Fe-Co DAzyme	OXD-like	TMB	0.47	1.833 × 10^–7^	------	[138]
	POD-like	H_2_O_2_	0.845	2.811 × 10^–7^	------	
	GSHOx-like	GSH	1.775	1.07 × 10^–6^	------	
Cu-JMCNs	POD-like	H_2_O_2_	0.37	4.6 × 10^–7^	------	[142]
S-N/Ni PSAE	POD-like	TMB	0.36	5.2× 10^–7^	------	[23]
		H_2_O_2_	0.082	4.2× 10^–7^	------	
IrN_5_ SAN	OXD-like	TMB	0.97	6.14 × 10^−8^	------	[25]
	POD-like	H_2_O_2_	2.43	9.23 × 10^−8^	------	
	NADHOD-like	NADH	24.28	1.76 × 10^–7^	------	
FeCo-DIA	POD-like	H_2_O_2_	0.122	8.37 × 10^−7^	------	[27]
FeNC-edge	POD-like	H_2_O_2_	71.2	1.91 × 10^−6^	1.33	[80]
		TMB	0.571	1.24 × 10^−6^	0.866	
	OXD-like	TMB	0.374	4.22 × 10^−8^	0.0295	
FeN_4_-SAzyme	POD-like	H_2_O_2_	285	6.20 × 10^−6^	37.83	[88]
		TMB	1.42	5.81 × 10^−8^	21.17	
	GSHOx-like	GSH	0.13	6.01 × 10^−7^	22.17	
Fe-CDs@Ang	OXD-like	TMB	0.071	1.71 × 10^−8^	------	[62]
	POD-like	H_2_O_2_	166	4.87 × 10^−7^	------	
		TMB	0.090	1.716 × 10^−7^	------	
	CAT-like	H_2_O_2_	0.12	1.718 × 10^−8^	------	
	GPx-like	H_2_O_2_	4.52	8.196 × 10^−5^	------	
Fe/PMCS SAzymes	OXD-like	TMB	0.0736	1.39 × 10^–8^	4.63 × 10^–2^	[100]
	POD-like	H_2_O_2_	3.54	3.30 × 10^–7^	1.10	
		TMB	0.0285	3.91 × 10^–7^	1.30	
Pd-C SAzyme	POD-like	H_2_O_2_	0.26	1.44 × 10^–8^	------	[96]
		TMB	0.638	3.25 × 10^–8^	------	
MoS_2_@SA-Fe	POD-like	H_2_O_2_	0.015	4.37×10^−8^	------	[77]
FeN_5_ SAzyme	POD-like	H_2_O_2_	11.2	2.96 × 10^–6^	1.37	[81]
		TMB	0.652	8.47 × 10^–6^	3.92	
ZnN_4_	POD-like	H_2_O_2_	0.31	4.14 × 10^−8^	------	[99]
PFCE@Co/A-TiO_2_ SAzyme	CAT-like	H_2_O_2_	1.95	2.96 × 10^–6^	------	[162]
Ru SAEs	GSHOx-like	GSH	5.43	5.55 × 10^–5^	------	[70]
	OXD-like	TMB	0.042	6.3 × 10^–7^	------	
	POD-like	H_2_O_2_	0.085	6.9 × 10^–7^	------	
		TMB	0.22	1.1 × 10^–6^	------	
MoSA–Nx–C	POD-like	TMB	0.79	1.12 × 10^−7^	0.06	[82]
		H_2_O_2_	2	3.7 × 10^−7^	0.2	
Pd SAzyme@PEG	POD-like	H_2_O_2_	1.79	1.51 × 10^−7^	------	[109]
	GSHOx-like	GSH	0.24	1.97 × 10^−5^	------	
Pd-Pta/Por	POD-like	H_2_O_2_	50.35	4.38 × 10^−8^	------	[140]
HA-NC Cu	POD-like	H_2_O_2_	103.89	5.26 × 10^−8^	------	[113]

## Figures and Tables

**Figure 1 ijms-24-15712-f001:**
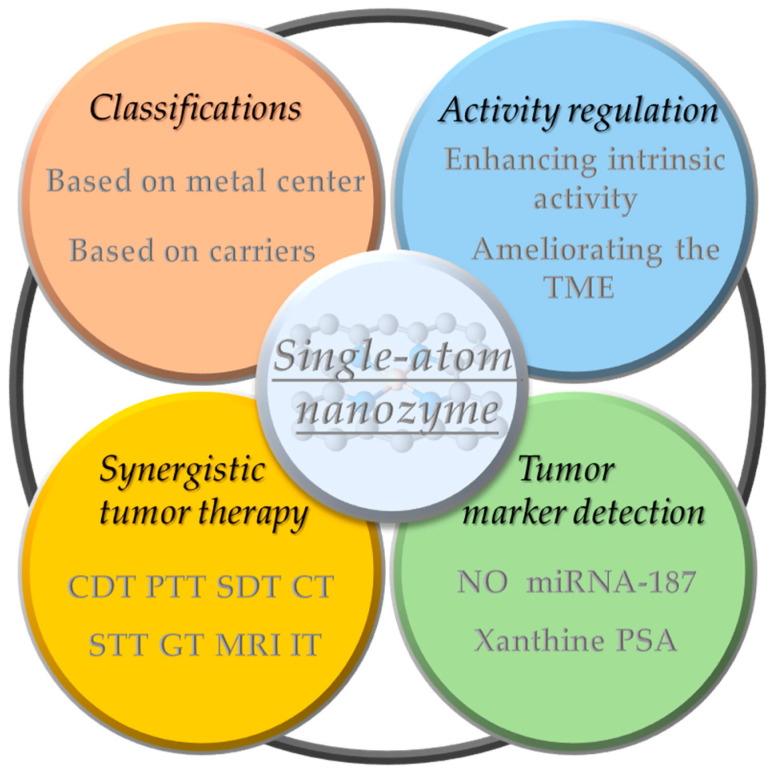
A brief overview of the classification, activity modulation, synergistic therapy, and marker detection of SANs applied to tumor diagnosis and therapy. (Tumor microenvironment (TME), chemodynamic therapy (CDT), photothermal therapy (PTT), sonodynamic therapy (SDT), chemotherapy (CT), sonothermal therapy (STT), gas therapy (GT), photodynamic therapy (PDT), magnetic resonance imaging (MRI), immunotherapy (IT), nitric oxide (NO), prostate-specific antigen (PSA)).

**Figure 3 ijms-24-15712-f003:**
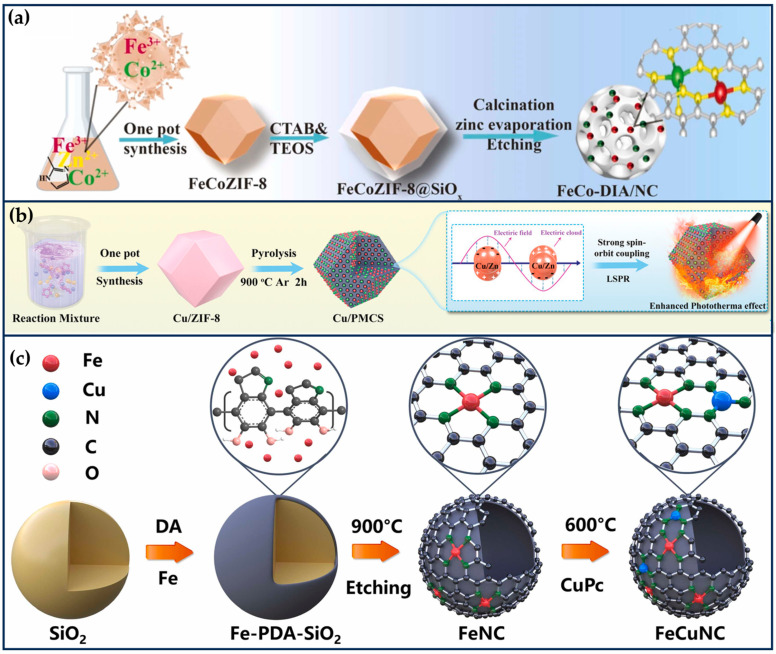
Dual-atom site nanozymes. (**a**) Synthetic schematic diagram of FeCo-DIA/NC (reproduced from [27] with permission from Elsevier). (**b**) The synthesis process of Cu/PMCS (reproduced from [54] with permission from John Wiley and Sons). (**c**) The preparation of FeCuNC (reproduced from [28] with permission from Elsevier).

**Figure 4 ijms-24-15712-f004:**
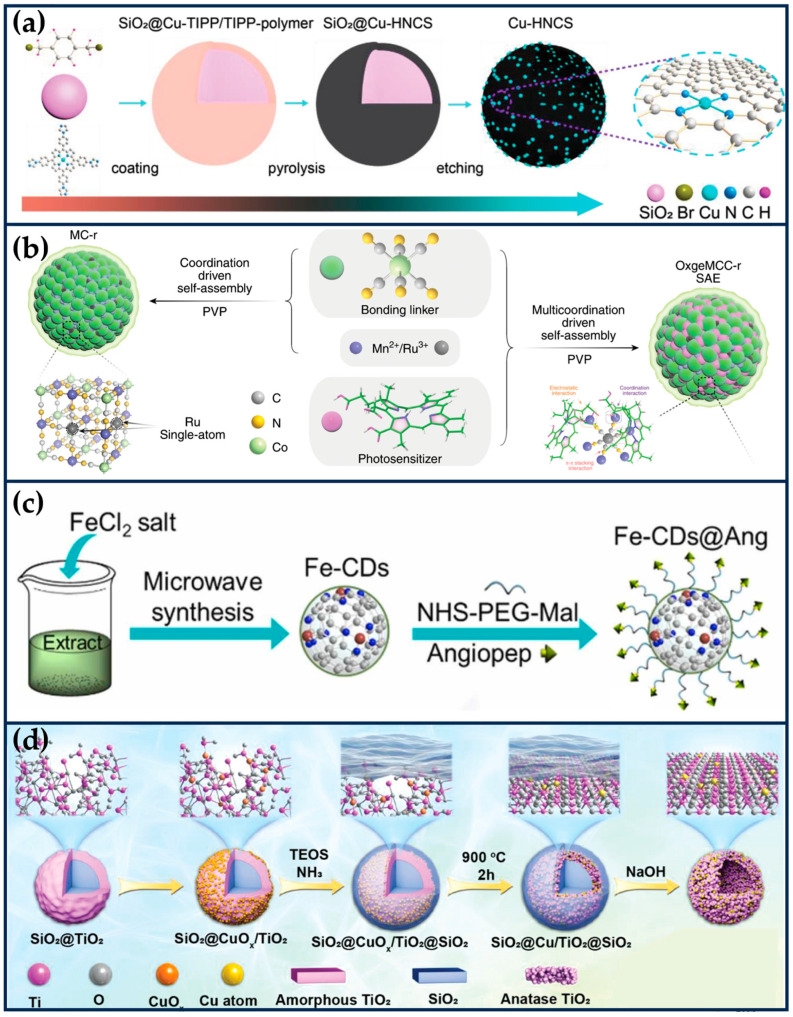
SANs with different support materials. (**a**) Schematic synthesis of Cu-HNCS by organic coating, carbonization, and silica etching (reproduced from [60] with permission from John Wiley and Sons). (**b**) OxgeMCC-r consists of catalytically active single-atom Ru site anchored in MCC with outer PVP protection layer (reproduced from [61] with permission from Springer Nature). (**c**) The preparation of Fe-CDs nanozyme (reproduced from [62] with permission from Elsevier). (**d**) The synthetic process of single-atom Cu/TiO2-PEG nanosonosensitizers by a reformative wrap–bake–strip method (reproduced from [63] with permission from John Wiley and Sons).

**Figure 5 ijms-24-15712-f005:**
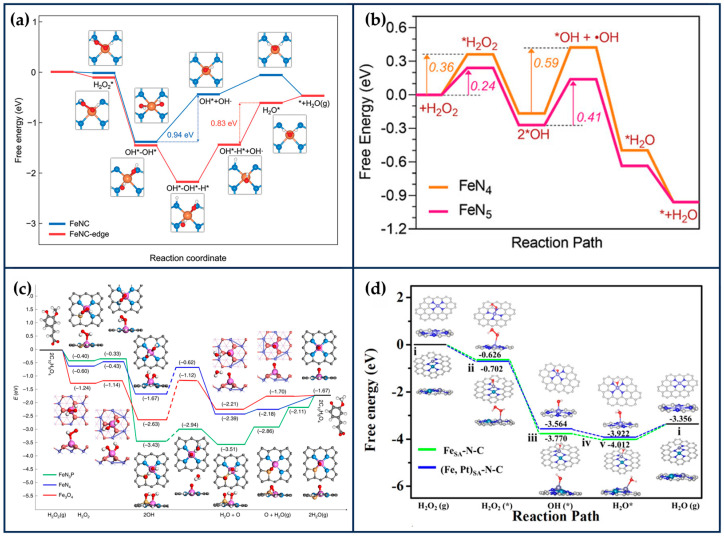
Theoretical calculations of reaction process. (**a**) Free energy diagram for POD-like reaction of FeNC and FeNC-edge (reproduced from [80] with permission from John Wiley and Sons). (**b**) Corresponding free energy diagram for peroxidase-like reaction on FeN_5_ and FeN_4_ (“*” represents intermediate) (reproduced from [81] with permission from John Wiley and Sons). (**c**) The energy profile diagram shows the most favorable paths of H_2_O_2_ dissociation into surface O species in neutral conditions, as well as the reaction 2C_10_H_9_O_4_  +  O → 2C_10_H_8_O_4_  +  H_2_O (g) (reproduced from [67] with permission from Springer Nature). (**d**) Free energy diagrams of (Fe, Pt)SA-N-C and FeSA-N-C in Fenton-like reaction (reproduced from [26] with permission from the American Chemical Society).

**Figure 6 ijms-24-15712-f006:**
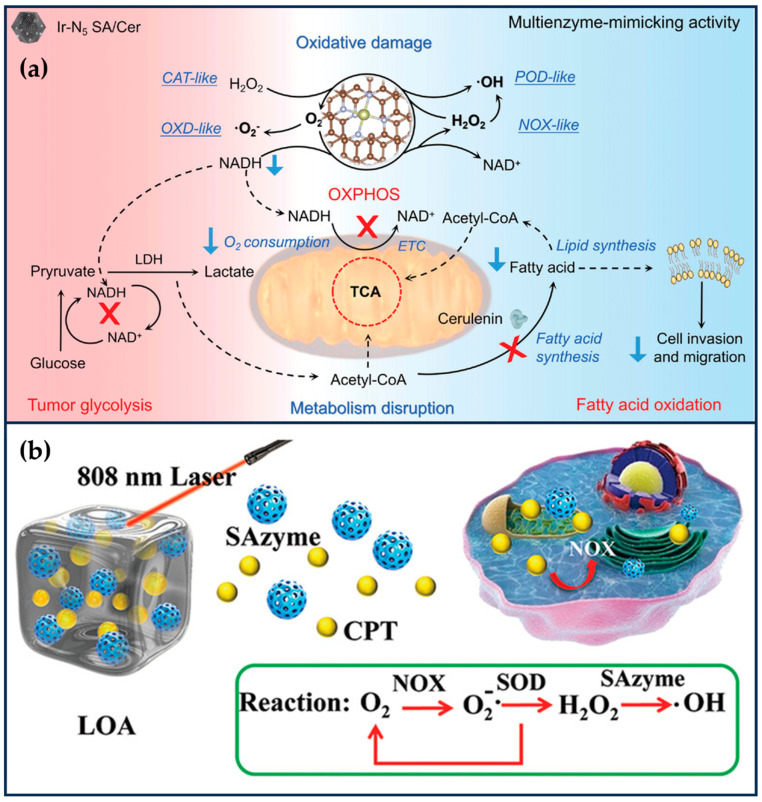
Ameliorating the TME. (**a**) Therapeutic mechanism of Ir N5 SA/Cer (reproduced from [25] with permission from John Wiley and Sons). (**b**) Schematic illustration of SAN-mediated •OH generation (reproduced from [96] with permission from John Wiley and Sons).

**Figure 7 ijms-24-15712-f007:**
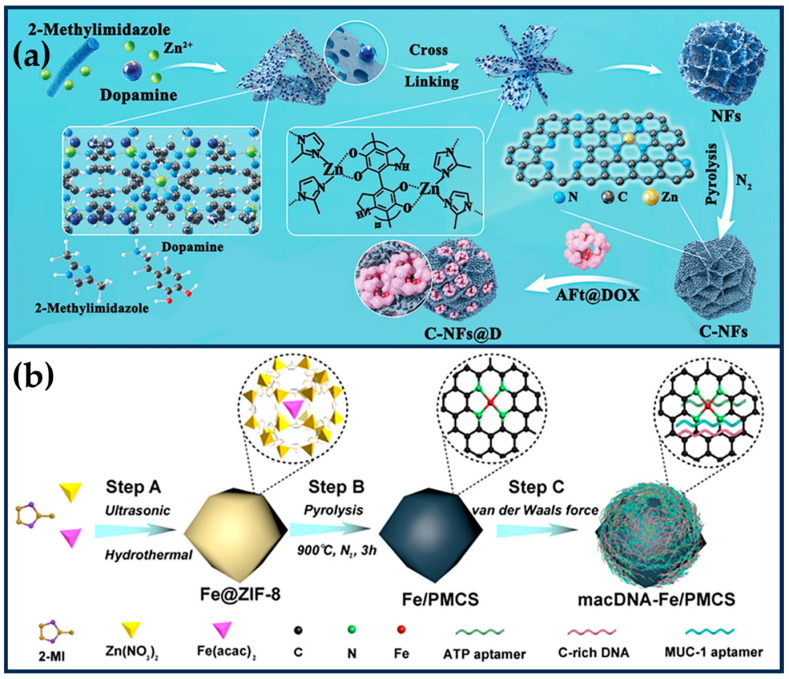
(**a**) Preparation process of MOF-derived SANs (C-NFs) with unique flower-like structures and further surface loading of AFt@DOX (reproduced from [99] with permission from the American Chemical Society). (**b**) Preparation of self-adaptive single-atom nanozymes of macDNA-Fe/PMCS (reproduced from [100] with permission from John Wiley and Sons).

**Figure 9 ijms-24-15712-f009:**
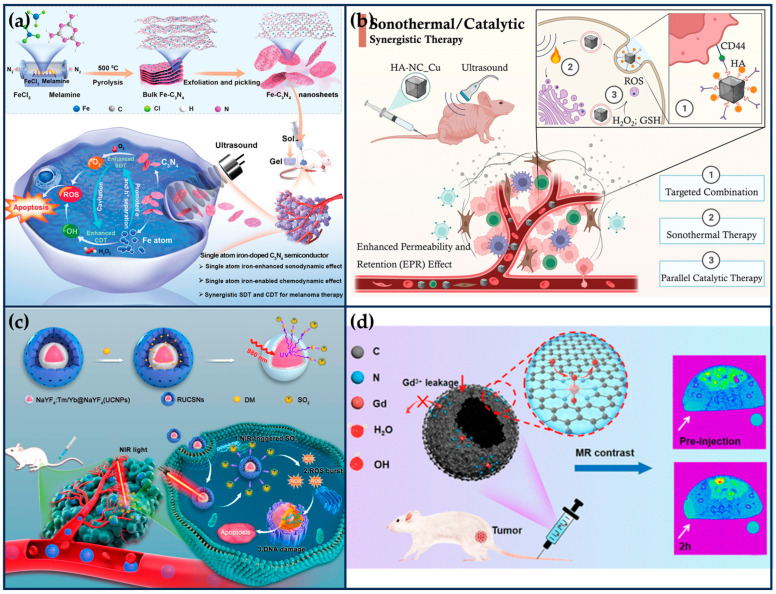
Synergistic tumor therapy with SANs in combination with other therapeutic modalities. (**a**) Schematic illustration of the synthesis of Fe- C_3_N_4_ NSs and the mechanism of SDT and CDT synergistic therapy by US-activated Fe-C_3_N_4_ (reproduced from [112] with permission from John Wiley and Sons). (**b**) Schematic illustration of HA-NC_Cu with excellent specificity to both endogenous TME and exogenous ultrasound irradiation for STT/CT synergistic treatment of triple-negative breast cancer (reproduced from [113] with permission from John Wiley and Sons). (**c**) Schematic illustration of the preparation process of RUCSNs-DM and intracellular SO2 generation and treatment under NIR light irradiation (reproduced from [114] with permission from the American Chemical Society). (**d**) Single-atom Gd nano-contrast agent enhances tumor MRI (reproduced from [115] with permission from the American Chemical Society).

## Data Availability

No new data were created or analyzed in this literature review. Data sharing is not applicable to this article.

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
