# Peer review of "Current Advances of Atomically Dispersed Metal-Centered Nanozymes for Tumor Diagnosis and Therapy"

_ijms, 2023, doi:10.3390/ijms242115712_

Round 1
Reviewer 1 Report
Comments and Suggestions for Authors
Dear authors,
The article provides a detailed review of the formulation of metal-centered Nanozymes for Tumor Diagnosis and Therapy. I consider the article to be of interest to the subject as it compiles extensive and up-to-date information. Many of the references belong to several groups that are repeatedly mentioned in the review. The figures are also obtained from the same cited references. However, the article presents an extensive, well-written, and understandable table where it compiles examples and reviews information regarding anticancer activity in tested models.
In mi opinion, the manuscript could be accepted after minor revision. I am attaching the manuscript with my comments and changes that I believe could improve the manuscript.

Author Response
Thank you very much for your efficient management of our manuscript (ijms-2626968) entitled “Current Advances of Atomically Dispersed Metal-centered Nanozymes for Tumor Diagnosis and Therapy”. We also appreciate the helpful comments from the reviewer. We have carefully revised the manuscript and provided the required experimental data according to the reviewer’s comments. A point-by-point response letter is provided. All the revised places are marked with yellow color.

Reviewer 2 Report
Comments and Suggestions for Authors
The review submitted by Tian et al. is a thick manuscript of 40 pages, including figures and a literature list. The whole manuscript is focused on the current advances of metal cation(s)-containing nanozymes in cancer therapy. The manuscript seems to be useful and could contribute to the field. However, its execution makes my numerous objections (listed below). Therefore, I cannot recommend publishing this review in the International Journal of Molecular Sciences.
Major issues:
1. My primary concern is related to the references used in this review. The reader might think that projects related to metal-centered nanozymes are mostly conducted in a single geographical region, which is not necessarily true. The authors should provide references from other groups from places.
2. In numerous places, the manuscript does not sound correct from the chemical and biochemical point of view. For instance, nanozymes do not contain metals but metal cations. Also, the authors rarely compare kinetic parameters between nanozymes; however, there is no comparison between described nanozymes and natural enzymes. The best parameter to compare nanozymes and enzymes is the catalytic proficiency index (kcat/Km). Another example is presented in Figure 2b, where BSA is modified and heated at 78 degrees C. This protein is denatured below 50 degrees C. On page 9 (line 363), the authors indicate that there are no chemotherapeutics against glioblastoma, which is not true (e.g., temozolomide)
3. The authors indicate that metal cation(s)-containing nanozymes are widely used in diagnosis and cancer therapy. Could the authors provide any example that is approved and used?
4. In some parts of the manuscript, the author focuses too much on chemical synthesis instead of biological activity
some corrections are required.
Author Response
Thank you very much for your efficient management of our manuscript (ijms-2626968) entitled “Current Advances of Atomically Dispersed Metal-centered Nanozymes for Tumor Diagnosis and Therapy”. We also appreciate the helpful comments from the reviewer. We have carefully revised the manuscript and provided the required experimental data according to the reviewer’s comments. A point-by-point response letter is provided. All the revised places are marked with yellow color.
- Suggestion 1: My primary concern is related to the references used in this review. The reader might think that projects related to metal-centered nanozymes are mostly conducted in a single geographical region, which is not necessarily true. The authors should provide references from other groups from places.
Response: Thanks for your suggestion. Firstly, in selecting these literatures we did not deliberately choose research results from a particular region, the selection process was based on the relevance of the articles to the topic of this review and the quality of the articles. The results of the selection show that many of the results are concentrated in a certain region, which indicates that researchers in this region have carried out a lot of explorations and published high-level articles in this field.
- Suggestion 2: In numerous places, the manuscript does not sound correct from the chemical and biochemical point of view. For instance, nanozymes do not contain metals but metal cations. Also, the authors rarely compare kinetic parameters between nanozymes; however, there is no comparison between described nanozymes and natural enzymes. The best parameter to compare nanozymes and enzymes is the catalytic proficiency index (kcat/Km). Another example is presented in Figure 2b, where BSA is modified and heated at 78 degrees C. This protein is denatured below 50 degrees C. On page 9 (line 363), the authors indicate that there are no chemotherapeutics against glioblastoma, which is not true (e.g., temozolomide)
Response: Thanks for your suggestion. Numerous international journals have pointed out that single-atom nanozymes have atomically dispersed metal centers. (Chem. Soc. Rev., 2022, 51, 3688-3734; Coord. Chem. Rev., 2020, 418, 213376; Adv. Mater., 2023, 2211724; ACS Cent. Sci., 2020, 6, 1288–1301; Sci. Adv., 2019, 5, eaav5490; Nat. Catal., 2021, 4, 407-417) We have added the relevant kinetic parameters in Table 2. It has been shown that the denaturation temperature of BSA can be increased with the involvement of metal ions (Food Hydrocoll., 2020, 101, 105450). In the example shown in Figure 2b, the stability of BSA may have been improved after coordination with Cu ions so that the complex of BSA with copper ions can remain stable at 78 °C. If the reviewer has any further questions, I suggest you email the original author of this literature to ask for relevant experimental details. Glioblastoma (GBM) is a fatal recurrent brain tumor for which there is no complete cure. The sentence on page 9 (line 363) has been corrected to read " Glioblastoma (GBM) is a fatal recurrent brain tumor for which there is no complete cure".
- Suggestion 3: The authors indicate that metal cation(s)-containing nanozymes are widely used in diagnosis and cancer therapy. Could the authors provide any example that is approved and used?
Response: Thanks for your suggestion. This article mainly wants to talk about the current development of single-atom nanozymes in tumor diagnosis and therapy. Although most of the single-atom nanozymes are still in the laboratory validation stage, they show promising in vivo and in vitro anti-tumor efficiencies, which makes us confident in the prospects of single-atom nanozymes and with further efforts to develop them, we believe that there will be single-atom nanozymes approved for use in the future as real anticancer drugs.
4. Suggestion 4: In some parts of the manuscript, the author focuses too much on chemical synthesis instead of biological activity.
Response: Thanks for your suggestion. We have described the chemical synthesis of many single-atom nanozymes in this review because we think it would be beneficial to assist readers in understanding how single-atom nanozymes are formed, what methods are used to construct single-atom nanozymes, and could potentially contribute to readers' own experiments.

Reviewer 3 Report
Comments and Suggestions for Authors
Comments to the author
The studies were well carried over, the figures and table were well structurally organized and arranged. I have few concerns and comments that need to be clarify before prior to publications. As a consequence, i recommend this manuscript can be consider as minor revision based on the following comments are attached.

Author Response
Thank you very much for your efficient management of our manuscript (ijms-2626968) entitled “Current Advances of Atomically Dispersed Metal-centered Nanozymes for Tumor Diagnosis and Therapy”. We also appreciate the helpful comments from the reviewer. We have carefully revised the manuscript and provided the required experimental data according to the reviewer’s comments. A point-by-point response letter is provided. All the revised places are marked with yellow color.
- Suggestion 1: If author could able to add few more additional information such as the as planned a detailed review could able to supporting their evidence for trapping free radicals by electron paramagnetic resonance (EPR)?? How this free radical is involved in cancer therapy diagnosis would also be very interesting to discuss. In addition, providing the band gap’s information’s are also increasing and correspondingly photo catalytic efficiency are also increasing! this has to be discussed further.
Response: Thanks for your suggestion. We have summarized in Table 1 the approaches to validating free radicals in the literatures. We briefly describe the catalytic therapeutic mechanism of single-atom nanozymes in Part 3. “Different types of SANs achieve the regulation of intracellular ROS levels mainly through POD-like, OXD-like, CAT-like, and GSHOx-like activities to kill cancer cells. Among them, the POD-like activities of SANs catalyze the generation of •OH with H2O2 as the electron acceptor when functioning. Due to the high reactivity of •OH, which can use various small molecules and macromolecules such as nucleic acids, lipids, and proteins as electron donors, causing severe oxidative damage to cancer cells. The CAT-like activity of SANs can decompose the endogenous H2O2 of the cancer cells into O2, which alleviates the anoxic environment of the tumor, thus increasing the efficiency of the generation of 1O2 in RT, PDT, and SDT. The OXD-like activity of SANs can catalyze the generation of H2O2 and , whose cancer cell-killing ability is weaker than that of •OH and needs to be combined with other enzyme-like activities to kill tumors. The GSHOx-like activity of SANs can consume excessive intracellular GSH, thus weakening the antioxidant ability of cancer cells and playing a very auxiliary therapeutic effect.” Suitable band gap values are indeed favorable for improving the photocatalytic efficiency of the catalysts. However, this review mainly outlines the application of single-atom nanozymes in tumor diagnosis and treatment and therefore does not discuss this aspect in detail.
- Suggestion 2: In discussion section, if author could able to discuss more details about Covalent organic frameworks (COF) based nanozyme for cancer therapy would also highly an encouraged, the most recent references are included in the revised? few recent studies are, Covalent Organic Framework Nanosheets as an Enhancer for Light Responsive Oxidase-Like Nanozymes: Multifunctional Applications in Colorimetric Sensing, Antibiotic Degradation, and Antibacterial Agents, ACS Sustainable Chemistry & Engineering, 2023, 11, 6956–6969.
Response: Thanks for your suggestion. As you mentioned, COF-based nanozymes are well employed in biology, and many of the COF-based single-atom catalysts show excellent performance for organic catalysis, hydrogen precipitation, and other fields. However, there are very few COF-based single-atom nanozymes applied in tumor therapy, so we have added the discussion of COF-based nanozymes and single-atom catalysts in the outlook part of the review, and we hope that in the future scientists will develop more high-performance COF-based single-atom nanozymes to be applied in tumor diagnosis and treatment.
- Suggestion 3: The Table, need to be incorporate more recent studies are which shows a better catalytic activity providing Vmax, Km and Kcat. I suggest author has to investigate with and without addition of recently developed nanozymes and compare their results in terms of oxidase-mimicking activity with providing catalytic parameter of Vmax, Km and Kcat, its very important to discuss further.
Response: Thanks for your suggestion. The main focus of this review is to provide an overview of the development of single-atom nanozymes for tumor diagnosis and treatment, and we have summarized high-level papers from the last few years. Kinetic parameters of the single-atom nanozymes have been added to Table 1.

Round 2
Reviewer 2 Report
Comments and Suggestions for Authors
The authors have addressed my comments therefore I suggest accepting the manuscript in its current form.
Comments on the Quality of English Languagethe language is fine.